# The cJUN NH$_2$-terminal kinase (JNK) signaling pathway promotes genome stability and prevents tumor initiation

Nomeda Girnius[1], Yvonne JK Edwards[1], David S Garlick[2], Roger J Davis[1,3]*

[1]Program in Molecular Medicine, University of Massachusetts Medical School, Worcester, United States; [2]Histo-Scientific Research Laboratories, Mount Jackson, United States; [3]Howard Hughes Medical Institute, University of Massachusetts Medical School, Worcester, United States

**Abstract** Breast cancer is the most commonly diagnosed malignancy in women. Analysis of breast cancer genomic DNA indicates frequent *loss-of-function* mutations in components of the cJUN NH$_2$-terminal kinase (JNK) signaling pathway. Since JNK signaling can promote cell proliferation by activating the AP1 transcription factor, this apparent association of reduced JNK signaling with tumor development was unexpected. We examined the effect of JNK deficiency in the murine breast epithelium. Loss of JNK signaling caused genomic instability and the development of breast cancer. Moreover, JNK deficiency caused widespread early neoplasia and rapid tumor formation in a murine model of breast cancer. This tumor suppressive function was not mediated by a role of JNK in the growth of established tumors, but by a requirement of JNK to prevent tumor initiation. Together, these data identify JNK pathway defects as 'driver' mutations that promote genome instability and tumor initiation.

DOI: https://doi.org/10.7554/eLife.36389.001

*For correspondence:
roger.davis@umassmed.edu

## Introduction

Breast cancer is the most frequently diagnosed tumor in women (*Siegel et al., 2015*). The etiology of breast cancer has been studied in detail, but the causes of breast cancer remain incompletely understood. Nevertheless, it is established that familial breast cancers result from germ-line mutations that increase the risk of cancer development (*Afghahi and Kurian, 2017*). Examples of inherited mutations that can cause breast cancer predisposition include *ATM*, *BRCA1/2*, *CDH1*, *CHEK2*, *NBN*, and *TP53*. Moreover, sporadic mutation of these and other genes promote the development of non-familial breast cancer (*Hanahan and Weinberg, 2011*). Changes in the tumor genome are therefore important for breast cancer development (*Hanahan and Weinberg, 2011*).

Recent advances in breast cancer genome analysis have led to significant progress towards the identification of sporadic mutations in breast cancer (*Cancer Genome Atlas Network, 2012*; *Ciriello et al., 2015*; *Ellis et al., 2012*; *Kan et al., 2010*; *Nik-Zainal et al., 2016*; *Shah et al., 2012*; *Stephens et al., 2012*; *Wang et al., 2014*). These genetic changes include 'driver' mutations that promote tumor development and 'passenger' mutations that do not functionally contribute to the tumor phenotype. Genes mutated in familial cancer syndromes constitute a prime example of 'driver' mutations that can contribute to cancer development. The presence of 'passenger' mutations complicates the analysis of cancer genomes for the development of targeted tumor therapy. For example, some 'driver' mutations cause genetic instability (e.g. *ATM*, *BRCA1/2*, *CHEK2*, *NBN* and *TP53*) that can result in the accumulation of additional mutations in developing tumors.

Computational methods have been employed to distinguish 'driver' and 'passenger' mutations based on mutation frequency (*Parmigiani et al., 2009*), gene function in pathways (*Lin et al., 2007*;

**eLife digest** As cells in our body grow and divide, their DNA can experience changes or damage. Most of these 'mutations' are harmless, or quickly fixed by the body. Yet, sometimes a mutation can trigger a chain of genetic events that drives the cells to multiply uncontrollably, which leads to tumors. Identifying these 'driver mutations' is complex, but key to understanding how cancers start and can be fought.

Breast cancer is the most common type of cancer diagnosed in women worldwide. Large studies have focused on sequencing the DNA of cancerous breast cells to try to identify the mutations that started the cancer. Results show that, in these cells, a biological mechanism called the JNK signaling pathway is often inactivated because mutations affect the molecules that take part in this process. Like a chain reaction, the proteins of the JNK pathway act on each other until the last one, called JNK, gets switched on. This protein then goes on to participate in a number of cellular processes such as DNA repair. Is it possible that mutations in this pathway actually drive cancer, and if so, how?

Girnius et al. addressed these questions by inactivating the JNK pathway in the breast cells of mice. Over the next year and a half, the JNK-deficient animals were more likely to get breast cancer than normal mice. Further experiments showed that, in breast cells, the JNK protein prevented tumors from appearing. However, once the tumors were present, it was less effective at stopping them from growing. The DNA of the breast cancer cells with no JNK protein also contained more genetic changes and mistakes. This suggests that the JNK signaling pathway helps to keep the genetic information 'healthy'. This may be because, normally, the JNK protein activates processes that fix DNA mutations. Taken together, the results presented by Girnius et al. show that genetic changes which inactivate the JNK pathway can drive the development of breast cancer.

Certain anti-cancer drugs kill cancerous cells by damaging their DNA. Breast tumor cells with inactive JNK pathways are less able to repair their genetic information, and so these drugs could potentially work well on them. Future experiments will be needed to test this hypothesis.

DOI: https://doi.org/10.7554/eLife.36389.002

---

*Wendl et al., 2011*), level of gene expression (*Berger et al., 2016*) and predictions based on gene function (*Carter et al., 2009*; *Kaminker et al., 2007*; *Youn and Simon, 2011*) and protein interactions (*Babaei et al., 2013*; *Cerami et al., 2010*). These computational approaches to identify 'driver' mutations have been complemented by functional siRNA screens on breast tumor cell lines (*Marcotte et al., 2016*; *Sanchez-Garcia et al., 2014*). Collectively, these approaches have led to the identification of 'driver' mutations in human cancer, but it is likely that many more 'driver' mutations remain to be discovered (*Garraway and Lander, 2013*).

Examples of 'driver' mutations in breast cancer include the *TP53*, *PIK3CA*, and *PTEN* genes. Mutational inactivation of *PTEN* or activation of *PI3K* increases AKT/mTOR signaling that promotes growth, proliferation, and survival (*Yuan and Cantley, 2008*), while mutation of *TP53* promotes cell survival and proliferation (*Vousden and Prives, 2009*). The appreciation of the importance of these pathways in cancer has spurred research into potential therapies (*Vousden and Prives, 2009*; *Yuan and Cantley, 2008*). These well-established 'driver' mutations contribute to the etiology of breast cancer. In contrast, the role of some other highly mutated genes in breast cancer is unclear.

One frequently mutated pathway in breast cancer is the cJUN $NH_2$-terminal kinase (JNK) pathway (*Garraway and Lander, 2013*). The JNK pathway is a three-tiered cascade that includes a MAP kinase kinase kinase (MAP3K) that phosphorylates and activates MAP kinase kinases (MAP2K) that, in turn, phosphorylate and activate JNK (*Davis, 2000*). This pathway requires two MAP2K isoforms that co-operate to activate JNK by phosphorylation on tyrosine (by MAP2K4) and threonine (by MAP2K7) (*Tournier et al., 2001*). The sequencing of breast tumor genomic DNA has revealed mutations in genes that encode members of this pathway, including *MAP3K1*, *MAP2K4*, and *MAP2K7* (*Banerji et al., 2012*; *Cancer Genome Atlas Network, 2012*; *Ciriello et al., 2015*; *Ellis et al., 2012*; *Kan et al., 2010*; *Nik-Zainal et al., 2016*; *Shah et al., 2012*; *Stephens et al., 2012*; *Wang et al., 2014*). The genetic changes include frequent deletion of the gene locus and mutations that cause protein truncation and loss of protein kinase activity. This analysis suggests that breast cancer is

associated with loss of JNK signaling. Indeed, since AKT phosphorylates and inactivates MAP2K4 (*Park et al., 2002*), breast cancer 'driver' mutations that activate AKT (e.g. *PTEN* and *PI3K*) also cause loss of JNK signaling. The JNK signaling pathway may therefore be suppressed in many breast cancers.

The association of breast cancer with reduced JNK signaling represents a correlation. What is the significance of *loss-of-function* JNK pathway mutations? Are these 'driver' or 'passenger' mutations? The purpose of this study was to test the role of JNK signaling in breast cancer development. Since JNK signaling causes AP1 transcription factor activation, we anticipated that JNK may act to promote tumor growth. In contrast, we found that loss of JNK signaling in mammary epithelial cells caused breast cancer. Furthermore, JNK deficiency accelerated tumor formation in a murine model of breast cancer. These effects of JNK deficiency to promote tumor development were associated with widespread presence of early neoplasia and genomic instability. We show that JNK plays a key role in the initiation of tumor development. Thus, the frequent *loss-of-function* JNK pathway mutations in breast tumors represent 'driver' mutations that promote breast cancer development.

## Results

### Disruption of JNK signaling causes breast cancer development

*Loss-of-function* mutations in the JNK signaling pathway (e.g. *MAP3K1*, *MAP2K4*, and *MAP2K7*) are implicated in the etiology of breast cancer (*Banerji et al., 2012*; *Cancer Genome Atlas Network, 2012*; *Ciriello et al., 2015*; *Ellis et al., 2012*; *Kan et al., 2010*; *Nik-Zainal et al., 2016*; *Shah et al., 2012*; *Stephens et al., 2012*; *Wang et al., 2014*). These potential 'driver' mutations in breast cancer cause disruption of JNK signaling. To test whether JNK pathway disruption influences breast cancer development, we examined the effect of JNK-deficiency in the mammary epithelium. The JNK1 (encoded by *Mapk8*) and JNK2 (encoded by *Mapk9*) isoforms exhibit partially redundant functions (*Davis, 2000*). We therefore examined compound JNK deficiency in the mammary epithelium using Control (ME$^{CRE}$: *Wap-Cre$^{+/-}$ Mapk8$^{+/+}$ Mapk9$^{+/+}$* and ME$^{WT}$: *Mapk8$^{LoxP/LoxP}$ Mapk9$^{LoxP/LoxP}$*) mice and JNK-deficient (ME$^{KO}$: *Wap-Cre$^{+/-}$ Mapk8$^{LoxP/LoxP}$ Mapk9$^{LoxP/LoxP}$*) mice. Lactation induces *Wap-Cre* expression (*Wagner et al., 1997*). Studies using *Rosa26$^{mTmG+/-}$* female reporter mice demonstrated *Cre*-mediated recombination in cytokeratin 8 (CK8) positive luminal epithelial cells (*Figure 1A*), but not in cytokeratin 5 (CK5) positive myoepithelial cells (*Figure 1B*).

We examined female control (ME$^{WT}$ and ME$^{CRE}$) mice and JNK-deficient (ME$^{KO}$) mice to determine whether JNK deficiency causes breast tumor development. We found no breast cancer or pre-malignant mammary lesions in a cohort of 22 control ME$^{CRE}$ mice (*Figure 1—source data 1*). However, studies of a cohort of 19 control ME$^{WT}$ mice identified one palpable breast tumor (adenosquamous carcinoma) in a 78 wk old mouse (*Figure 1C*). In addition, mammary intraepithelial neoplasia (MIN) was detected in one ME$^{WT}$ mouse during microscopic analysis of tissue sections following necropsy. In contrast, the incidence of palpable breast tumors (31% of mice, median age 80 wk; p=0.037; Fisher's Exact Test) and MIN lesions detected at necropsy (41% of mice; p=0.0084; Fisher's Exact Test) in a cohort of 32 ME$^{KO}$ mice was significantly greater than ME$^{WT}$ mice (*Figure 1C*). Immunoblot analysis confirmed that the ME$^{KO}$ adenocarcinoma and adenosquamous carcinoma cells do not express JNK proteins (*Figure 1D*). These data indicate that JNK deficiency promotes breast tumor development.

Microscopic analysis of tumor sections demonstrated the presence of adenosquamous carcinoma in ME$^{WT}$ mice, but both adenocarcinoma and adenosquamous carcinoma were detected in ME$^{KO}$ mice (*Figure 1C,E,F*). The adenocarcinomas were primarily CK8 positive and variably expressed estrogen receptor (ER) and progesterone receptor (PR), while the adenosquamous carcinomas expressed CK5 and did not express ER (*Figure 1—figure supplement 1A,B*). CK8 expression by ME$^{KO}$ adenocarcinomas is consistent with both a luminal epithelial cell origin and the expression of *Wap-Cre* in luminal epithelial cells (*Figure 1A,B*). In contrast, CK5 expression by the ME$^{KO}$ adenosquamous carcinomas suggests that these tumor cells may partially differentiate to express a marker of myoepithelial cells.

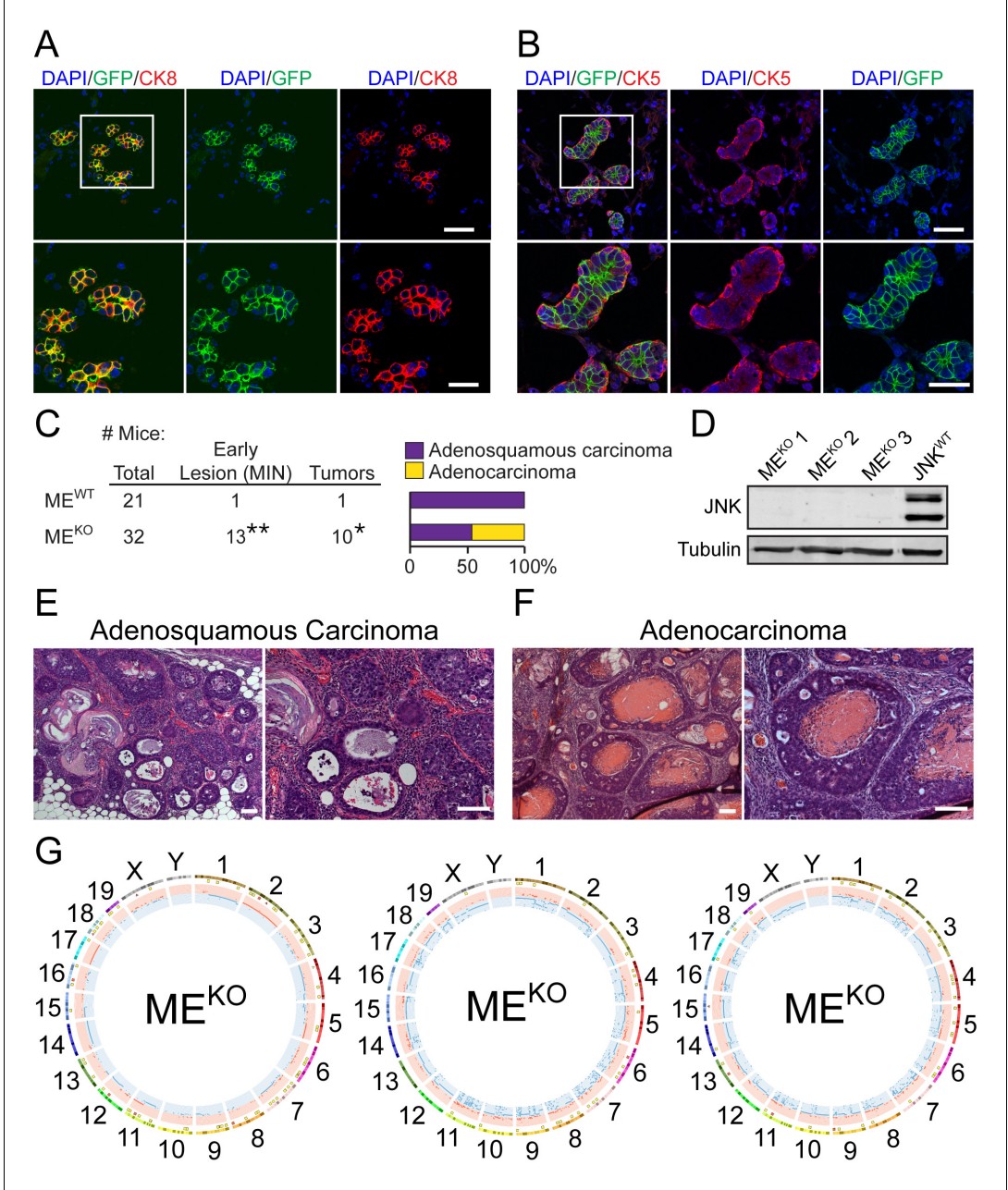

**Figure 1.** JNK deficiency in mammary epithelial cells causes genomic instability and tumor formation. (**A, B**) Mammary gland tissue sections were prepared from parous *Wap-Cre^{+/-} Rosa26^{mTmG+/-}* female mice (n = 6) on day 21 post-weaning. These sections were stained with antibodies to cytokeratin 8 (CK8, red (**A**)) or cytokeratin 5 (CK5, red (**B**)), and GFP (green), and counterstained with DAPI (blue). Representative images are presented (*upper panel*, scale bar = 48 µm). Boxed area was magnified (*lower panel*, scale bar = 24 µm). (**C**) Summary of the study cohort showing the total number of *Mapk8^{LoxP/LoxP} Mapk9^{LoxP/LoxP}* (ME^{WT}) and *Wap-Cre^{+/-} Mapk8^{LoxP/LoxP} Mapk9^{LoxP/LoxP}* (ME^{KO}) mice examined, the number of mice exhibiting mammary intraepithelial neoplasia (MIN) or tumors (*p=0.037, **p=0.0084; Fisher's Exact Test) (*left panel*). The type of carcinoma is presented (*right panel*). (**D**) Extracts prepared from ME^{KO} and *Wap-Cre^{+/-} Trp53^{LoxP/LoxP}* (JNK^{WT}) tumor cells were subjected to immunoblot analysis using antibodies to JNK and α-Tubulin. The ME^{KO} tumors examined were representative of adenocarcinoma (#1), tumors with characteristics of both adenocarcinoma and adenosquamous carcinoma (#2), and adenosquamous carcinoma (#3). (**E, F**) Representative hematoxylin and eosin (H and E) -stained sections of adenosquamous carcinomas (**E**) and adenocarcinomas (**F**) from ME^{KO} female mice are presented. Scale bar = 100 µm. (**G**) Exome sequencing was performed on ME^{KO} tumor cell lines (n = 3). Mammary tissue from a virgin female of the same genotype (*Wap-Cre^{+/-} Mapk8^{LoxP/LoxP} Mapk9^{LoxP/LoxP}*) was used as the reference genome. Circos plots showing copy number variations (CNVs) in ME^{KO} tumor cells are presented. The outermost ring shows chromosome ideograms. The next track indicates high (red) and moderate (yellow) impact single nucleotide variants and indels marked by rectangles

*Figure 1 continued on next page*

*Figure 1 continued*

and triangles, respectively. The innermost track shows chromosome amplifications and deletions, with red and blue lines indicating chromosomal fragments present at $\log_2$(ratio tumor/normal)>0.2 or $\log_2$(ratio tumor/normal)<−0.2, respectively.

DOI: https://doi.org/10.7554/eLife.36389.003

The following source data and figure supplements are available for figure 1:

**Source data 1.** Spreadsheet of source data for *Figure 1C*.

DOI: https://doi.org/10.7554/eLife.36389.006

**Source data 2.** Source image data for *Figure 1D*.

DOI: https://doi.org/10.7554/eLife.36389.007

**Figure supplement 1.** Expression of estrogen and progesterone receptors in breast tumors caused by JNK deficiency in the mammary epithelium.

DOI: https://doi.org/10.7554/eLife.36389.004

**Figure supplement 2.** Summary of exome sequence data.

DOI: https://doi.org/10.7554/eLife.36389.005

## JNK deficiency promotes genomic instability and altered gene expression

The JNK pathway has been implicated in genome maintenance (*Calses et al., 2017*; *Lu et al., 2006*; *Van Meter et al., 2016*). To test whether JNK deficiency caused defects in genome stability, we examined the exome sequences of three independent ME$^{KO}$ breast tumor cell lines (*Figure 1D* and *Figure 1—figure supplement 2A*). This analysis demonstrated single nucleotide variants (SNVs) and short insertions/deletions (Indels) (*Figure 1G*), including high impact SNVs and Indels (frame-shifts and stop codons) within gene coding regions (*Figure 1—figure supplement 2B,C*). Moreover, many chromosome segment amplifications and deletions (copy number variations, CNVs) were detected (*Figure 1G*). These data demonstrate that genomic instability is a consequence of JNK deficiency.

We compared gene expression profiles of three independent ME$^{KO}$ tumor-derived cell lines and three independent primary *Wap-Cre$^{+/-}$* epithelial cell (MEC) preparations. RNA-seq analysis identified 2217 differentially expressed genes (q < 0.05, |$\log_2$ Fold Change| > 0.75) that formed two clusters (*Figure 2A*). Pathway over-representation analysis using the Kyoto Encyclopedia of Genes and Genomes (KEGG) database revealed that both clusters were highly enriched for 'Pathways in Cancer' (*Figure 2B*). Cluster one was also enriched for 'Focal Adhesion', while Cluster two was enriched for 'Metabolic Pathways' and 'p53 Signaling' (*Figure 2B*). As expected, JNK deficiency caused reduced expression of AP1 transcription factors (*Ventura et al., 2003*) (*Figure 2C*). We also observed reduced expression of a 'DNA repair' gene signature (*Figure 2—figure supplement 1*) in ME$^{KO}$ tumor cells compared with primary mammary epithelial cells (MEC), consistent with the detection of accumulated mutations in ME$^{KO}$ tumor cells (*Figure 1G*).

Ingenuity Pathway Analysis (IPA) identified a significant increase in 'WNT/β-Catenin Pathway' activity in ME$^{KO}$ tumor cells compared with MEC (*Figure 2D*). Moreover, an enrichment of WNT signaling genes was found in the pathway over-representation analysis (*Figure 2B*, Cluster 2), including increased expression of *Wnt7b* and *Wnt10a* (*Figure 2E*) and increased expression of the WNT target genes *Axin2*, *Ccnd1*, and *Myc* in ME$^{KO}$ cells compared with MEC (*Figure 2F*). These data suggest that ME$^{KO}$ tumors may exploit the WNT pathway during tumor development.

## JNK deficiency rapidly accelerates tumor development in a mouse model of breast cancer

The observation that JNK deficiency promotes breast tumorigenesis (*Figure 1*) suggests that defects in JNK signaling may accelerate tumor development in a sensitized genetic background. To test this hypothesis, we examined the genetic interaction between JNK inactivation and loss of TRP53. Since *TP53* is the most frequently mutated gene in human breast cancer, this model is relevant to human disease (*Nik-Zainal et al., 2016*).

We established TRP53-deficient mice (JNK$^{WT}$: *Wap-Cre$^{+/-}$ Trp53$^{LoxP/LoxP}$*) and TRP53/JNK compound mutant mice (JNK$^{KO}$: *Wap-Cre$^{+/-}$ Trp53$^{LoxP/LoxP}$ Mapk8$^{LoxP/LoxP}$ Mapk9$^{LoxP/LoxP}$*). Loss of JNK on the TRP53-deficient background dramatically accelerated tumor formation (*Figure 3A*). Histological examination of the JNK$^{WT}$ and JNK$^{KO}$ tumors confirmed that the majority of lesions were adenocarcinomas (*Figure 3B* and *Figure 3—figure supplement 1A*). JNK$^{WT}$ and JNK$^{KO}$ tumors presented

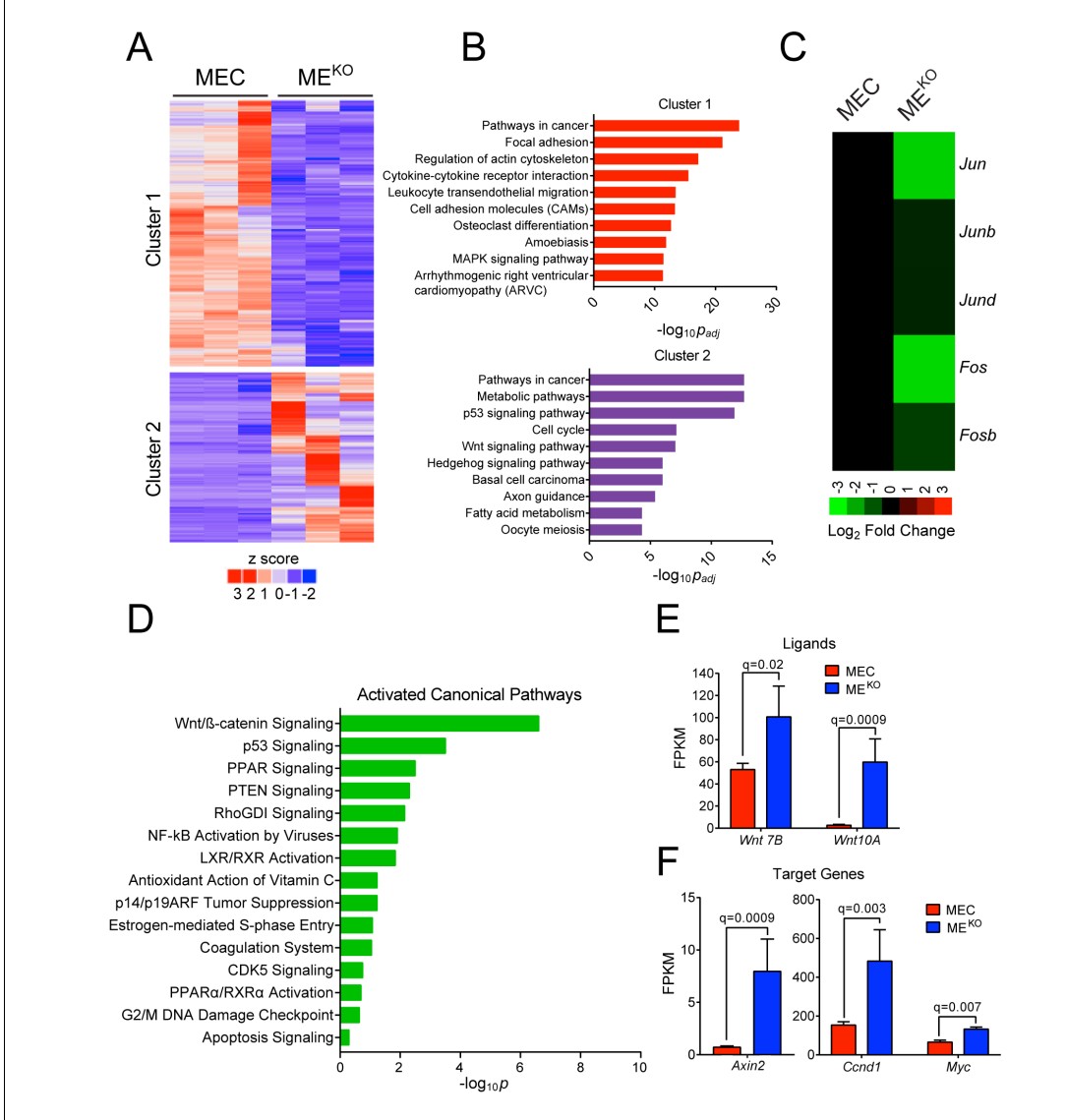

**Figure 2.** JNK deficiency promotes tumor-associated gene expression. (A–D) RNA-seq analysis was performed using primary mammary epithelial cells (MEC, n = 3) and ME$^{KO}$ tumor cell lines (n = 3). K-means clustering was performed on differentially expressed genes and is presented as a heatmap (A). Pathway over-representation analysis using the KEGG database was performed on genes from each of the clusters. The pathways with the 10 lowest p$_{adj}$ values are presented (B). The mean expression *Jun*, *Junb*, *Jund*, *Fos*, and *Fosb* mRNA is presented as a heatmap (C). Ingenuity Pathway Analysis of the RNA-seq data was used to predict signaling pathway activity (D). (E) *Wnt7b* and *Wnt10a* expression in MEC (n = 3) and ME$^{KO}$ tumor cells (n = 3) is presented as the mean fragments per kilobase of exon model per million mapped fragments (FPKM) ± SEM. (F) WNT target gene expression (*Axin2*, *Ccnd1*, and *Myc*) in MEC (n = 3) and ME$^{KO}$ (n = 3) cells is presented as the mean FPKM ± SEM.

DOI: https://doi.org/10.7554/eLife.36389.008

The following source data and figure supplement are available for figure 2:

**Source data 1.** Spreadsheet of source data for *Figure 2*.
DOI: https://doi.org/10.7554/eLife.36389.010

**Figure supplement 1.** Gene set enrichment analysis demonstrates that JNK deficiency causes decreased expression of a 'DNA Repair' gene signature.
DOI: https://doi.org/10.7554/eLife.36389.009

with variable ER and PR staining patterns (*Figure 3C* and *Figure 3—figure supplement 1B*). Moreover, both JNK$^{WT}$ and JNK$^{KO}$ tumor cells primarily expressed the luminal marker CK8, consistent with a luminal epithelial cell origin, although some dispersed cells did express the myoepithelial cell marker CK5 (*Figure 3C*).

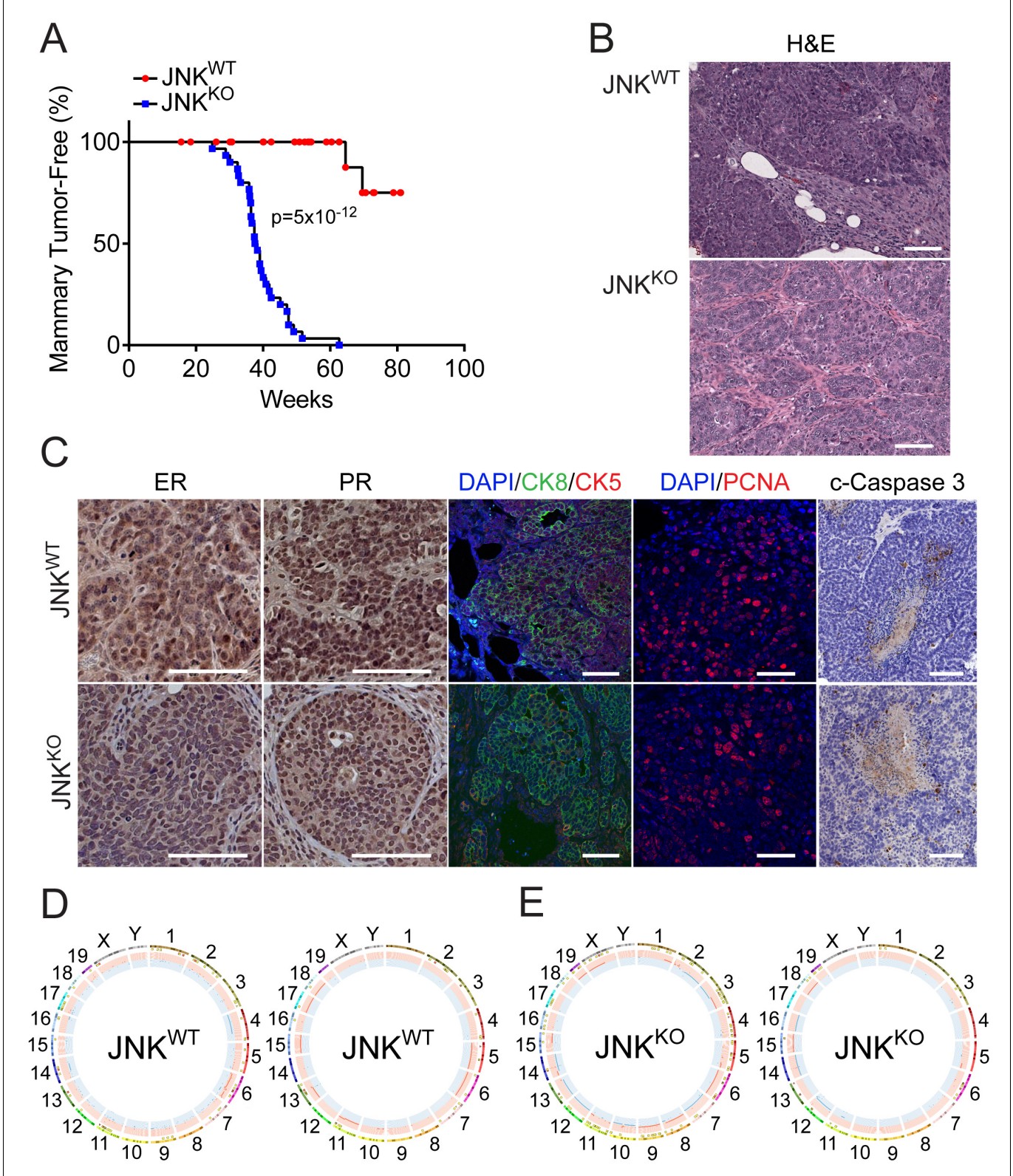

**Figure 3.** JNK deficiency accelerates tumor formation in a mouse model of breast cancer. (**A**) Mammary tumor-free survival was monitored in cohorts of 26 *Wap-Cre*[+/−] *Trp53*[LoxP/LoxP] (JNK[WT]) mice and 32 *Wap-Cre*[+/−] *Trp53*[LoxP/LoxP] *Mapk8*[LoxP/LoxP] *Mapk9*[LoxP/LoxP] (JNK[KO]) mice. Animals euthanized before a palpable mammary tumor had formed were censored in the log-rank analysis. (**B**) Tissue sections were prepared from JNK[WT] mammary tumors (n = 11) and JNK[KO] mammary tumors (n = 35). Representative images of H and E-stained sections from JNK[WT] (*upper panel*) and JNK[KO] (*lower panel*)
*Figure 3 continued on next page*

Figure 3 continued

mice are presented. Scale bar = 100 µm. (C) Adenocarcinoma tissue sections from JNK<sup>WT</sup> mice (*upper panel*) and JNK<sup>KO</sup> mice (*lower panel*) were stained with antibodies to (from left to right) estrogen receptor (ER), progesterone receptor (PR), cytokeratins 5 (red) and 8 (green) (CK5 and CK8 respectively), PCNA (Scale bars = 50 µm), and cleaved caspase 3 (Scale bar = 100 µm). Immunofluorescent stains were counterstained with DAPI, and peroxidase-based staining was counterstained with hematoxylin. Representative images are presented. (D, E) Exome sequencing was performed on JNK<sup>WT</sup> (n = 2) and JNK<sup>KO</sup> (n = 6) tumor cell lines. Mammary tissue from a virgin female of the same genotype (*Wap-Cre^{+/-} Trp53^{LoxP/LoxP}* for JNK<sup>WT</sup> and *Wap-Cre^{+/-} Trp53^{LoxP/LoxP} Mapk8^{LoxP/LoxP} Mapk9^{LoxP/LoxP}* for JNK<sup>KO</sup>) was used as the reference genome. Representative Circos plots showing CNVs are presented for JNK<sup>WT</sup> (D) and JNK<sup>KO</sup> (E) tumor cells. The outermost ring shows chromosome ideograms. The next track indicates high (red) and moderate (yellow) impact single nucleotide variants and indels marked by rectangles and triangles, respectively. The innermost track shows chromosome amplifications and deletions, with red and blue lines indicating chromosomal fragments present at $\log_2$(ratio tumor/normal)>0.2 or $\log_2$(ratio tumor/normal)<−0.2, respectively.
DOI: https://doi.org/10.7554/eLife.36389.011

The following source data and figure supplements are available for figure 3:

**Source data 1.** Spreadsheet of source data for *Figure 3A*.
DOI: https://doi.org/10.7554/eLife.36389.015
**Figure supplement 1.** Tumors in JNK<sup>KO</sup> mice are primarily adenocarcinomas and display a spectrum of hormone receptor expression patterns.
DOI: https://doi.org/10.7554/eLife.36389.012
**Figure supplement 2.** Exome sequencing of Control and JNK-deficient tumor cells.
DOI: https://doi.org/10.7554/eLife.36389.013
**Figure supplement 2—source data 1.** Spreadsheet of source data for *Figure 3—figure supplement 2C*.
DOI: https://doi.org/10.7554/eLife.36389.016
**Figure supplement 3.** Gene set enrichment analysis demonstrates increased expression of a 'KRAS Signaling' gene signature in breast tumor cells.
DOI: https://doi.org/10.7554/eLife.36389.014

To test whether JNK deficiency promotes tumorigenesis by disrupting the balance of proliferation and cell death, we stained tumor tissue sections with antibodies to detect the proliferation marker PCNA and the apoptotic marker cleaved caspase 3. No differences were detected in PCNA-stained sections, indicating that JNK<sup>KO</sup> tumors were not more proliferative than JNK<sup>WT</sup> tumors (*Figure 3C*). Similarly, we did not detect differences in cleaved caspase 3 staining between JNK<sup>WT</sup> and JNK<sup>KO</sup> tumors (*Figure 3C*). Collectively, these data show that JNK deficiency causes significantly accelerated disease progression without greatly changing the tumor phenotype.

To assess whether there are differences in mutational load, and to determine if there are recurring mutations or chromosomal alterations associated with the different tumor genotypes, we performed exome sequencing of JNK<sup>WT</sup> and JNK<sup>KO</sup> tumor cell lines (*Figure 3D,E* and *Figure 1—figure supplement 2A*). High impact SNVs and Indels were identified (*Figure 1—figure supplement 2B, C*). CNV analysis demonstrated that chromosome six was amplified and no chromosome was consistently deleted in JNK<sup>WT</sup> tumor cells (*Figure 3D*). Chromosome six was also amplified in some (two of six) JNK<sup>KO</sup> tumor cells (*Figure 3E* and *Figure 3—figure supplement 2A*). The proto-oncogene *Kras* resides on chromosome six and CNV analysis demonstrated that the *Kras* locus was recurrently amplified in JNK<sup>WT</sup> tumor cells (*Figure 3D* and *Figure 3—figure supplement 2A*) and was more highly expressed (*Figure 3—figure supplement 2B*). The reduced amplification of the *Kras* locus in JNK<sup>KO</sup> tumor cells compared with JNK<sup>WT</sup> tumor cells suggests that an alternative mechanism of *Kras* regulation may contribute to the phenotype of these tumor cells. Indeed, increased expression of a 'KRAS signaling' gene signature was detected in both JNK<sup>WT</sup> and JNK<sup>KO</sup> tumor cells (*Figure 3—figure supplement 3*).

## Effects of JNK deficiency on tumor-associated gene expression

We performed RNA-seq on primary mammary epithelial cells (MEC) and tumor cells (*Figure 4—figure supplement 1A*) to test whether JNK deficiency caused changes in tumor-associated gene expression. An examination of genes differentially expressed ($|\log_2$ Fold Change| > 0.75; q < 0.05) in tumor cells compared to MEC revealed distinct gene expression patterns between JNK<sup>WT</sup>, JNK<sup>KO</sup>, and ME<sup>KO</sup> tumor cells (*Figure 4A* and *Figure 4—figure supplement 1B*). Pathway over-representation analysis using the KEGG database (*Figure 4B*) demonstrated that up-regulated genes in JNK-deficient tumors were enriched for 'Metabolic Pathways' (Clusters 1 and 2), while genes up-regulated in JNK<sup>WT</sup> tumors were enriched for 'Cytokine-Cytokine Receptor Interactions' (Cluster 3). In

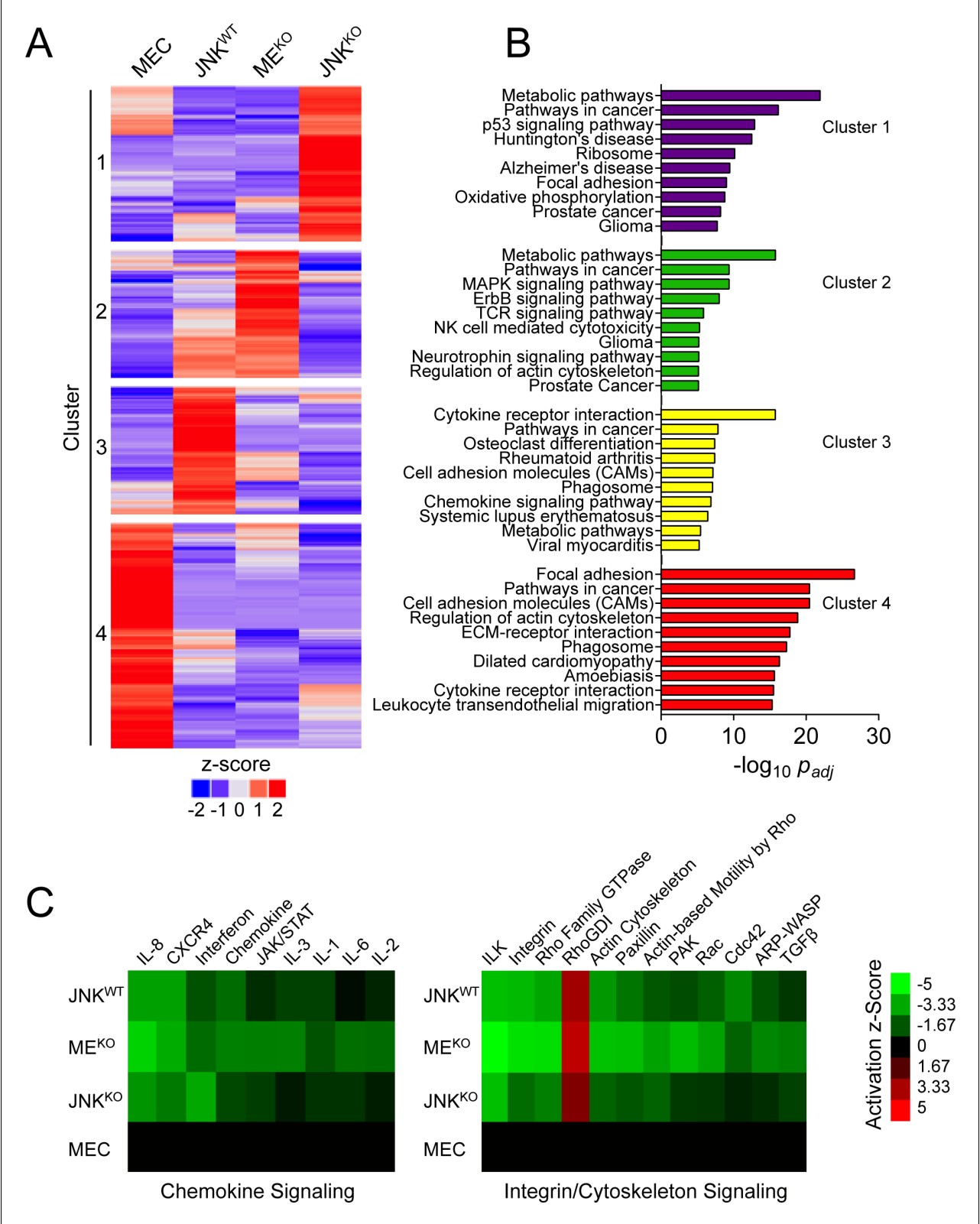

**Figure 4.** RNA-seq analysis demonstrates that a sub-set of tumor-associated gene expression requires JNK. (A, B) RNA isolated from primary mammary epithelial cells (MEC, n = 3) and also JNK$^{WT}$ (n = 2), ME$^{KO}$ (n = 3), and JNK$^{KO}$ (n = 2) tumor cell lines was sequenced. The heatmap presents k-means clustering (k = 4) of genes differentially expressed in any of the pairwise comparisons (q < 0.05, |log$_2$ Fold Change| > 0.75; mean) (A). Pathway over-representation analysis was performed on each of the four clusters using the KEGG database (B). The pathways with the lowest $p_{adj}$ values for each

*Figure 4 continued on next page*

*Figure 4 continued*

cluster are presented. (**C**) Comparative analysis was performed on genes differentially expressed between MEC and the tumor cell lines using Ingenuity Pathway Analysis (IPA). Heatmaps show the predicted activation (Activation z-score) of canonical pathways involved in immune (*left panel*) and integrin/cytoskeleton (*right panel*) signaling (cutoff score = 1.31; equates to p=0.049 using Fisher's Exact Test).

DOI: https://doi.org/10.7554/eLife.36389.017

The following source data and figure supplements are available for figure 4:

**Source data 1.** Spreadsheet of source data for *Figure 4*.

DOI: https://doi.org/10.7554/eLife.36389.020

**Figure supplement 1.** Gene expression analysis of control and JNK-deficient tumor cells.

DOI: https://doi.org/10.7554/eLife.36389.018

**Figure supplement 1—source data 1.** Spreadsheet of source data for *Figure 4—figure supplement 1*.

DOI: https://doi.org/10.7554/eLife.36389.021

**Figure supplement 2.** Comparison of signaling pathway activity in control and JNK-deficient tumor cells.

DOI: https://doi.org/10.7554/eLife.36389.019

**Figure supplement 2—source data 1.** Source image data for *Figure 4—figure supplement 2A*.

DOI: https://doi.org/10.7554/eLife.36389.022

**Figure supplement 2—source data 2.** Source image data for *Figure 4—figure supplement 2B*.

DOI: https://doi.org/10.7554/eLife.36389.023

contrast, down-regulated genes in all of the tumor cells were enriched for 'Focal Adhesion Proteins' (Cluster 4). These data indicate that JNK deficiency selectively alters a subset of tumor-associated gene expression.

To understand how JNK deficiency may alter tumor-signaling pathways, we used IPA to predict pathway activation status by examining differential expression between each tumor type and MEC. The pathways were ranked by Activation z-Score (total -$\log_{10}p$ of Fisher's Exact Test across the tumors) and the top 100 were considered (*Figure 4—figure supplement 1C*). Two dominant categories emerged from the comparative analysis: down-regulated 'Chemokine/Cytokine Signaling'; and down-regulated 'Integrin/Cytoskeleton Signaling' (*Figure 4C*).

We also examined signaling pathways by immunoblot analysis. Studies of JNK$^{WT}$ tumor cells demonstrated the presence of a functional JNK signaling pathway, including stress-induced phosphorylation of both JNK and cJUN (*Figure 4—figure supplement 2A*). However, JNK was not detected in JNK$^{KO}$ tumor cells (*Figure 4—figure supplement 2B*). The activation state of other MAPK pathways (ERK and p38) and the AKT pathway were similar between JNK$^{WT}$ and JNK$^{KO}$ tumor cells (*Figure 4—figure supplement 2B*).

## JNK deficiency does not increase tumor stem cell activity

The increased mammary tumor formation (*Figure 3A*) caused by JNK deficiency may reflect a role of JNK in tumor stem cells. To examine this potential role of JNK, we monitored mammosphere formation and maintenance using JNK$^{WT}$ and JNK$^{KO}$ tumor cells. No evidence of increased mammosphere propagation by the JNK$^{KO}$ tumor cells was obtained (*Figure 5A,B*). Moreover, we did not detect enhanced sphere formation by ME$^{KO}$ tumor cells (*Figure 5—figure supplement 1A,B*). Staining of agarose-embedded JNK$^{WT}$ and JNK$^{KO}$ mammospheres revealed a similar organization with peripheral CK5$^+$ cells and central CK8$^+$ cells (*Figure 5C*). Finally, the expression of stem cell markers (*Bmi1*, *Nanog*, and *Pou5f1*) was not significantly different between JNK$^{WT}$ and JNK$^{KO}$ mammospheres (*Figure 5D*). This analysis does not support the conclusion that differences in cancer stem cell activity account for the accelerated tumor formation by JNK$^{KO}$ mice compared with JNK$^{WT}$ mice.

## JNK deficiency does not increase tumor cell proliferation, but does promote survival

Transformed cells co-opt cellular processes to block anti-proliferative and death mechanisms while increasing cell proliferation, invasion, and migration (*Hanahan and Weinberg, 2011*). A change in one of these processes may account for the increased tumor formation detected in JNK$^{KO}$ mice. No differences between JNK$^{WT}$ and JNK$^{KO}$ tumor cell proliferation were detected (*Figure 6A*). Similarly, we found no difference in JNK$^{WT}$ and JNK$^{KO}$ tumor cell migration during wound healing (*Figure 6B*) or in response to a serum gradient (*Figure 6C*). Moreover, JNK$^{WT}$ and JNK$^{KO}$ tumor cell invasion

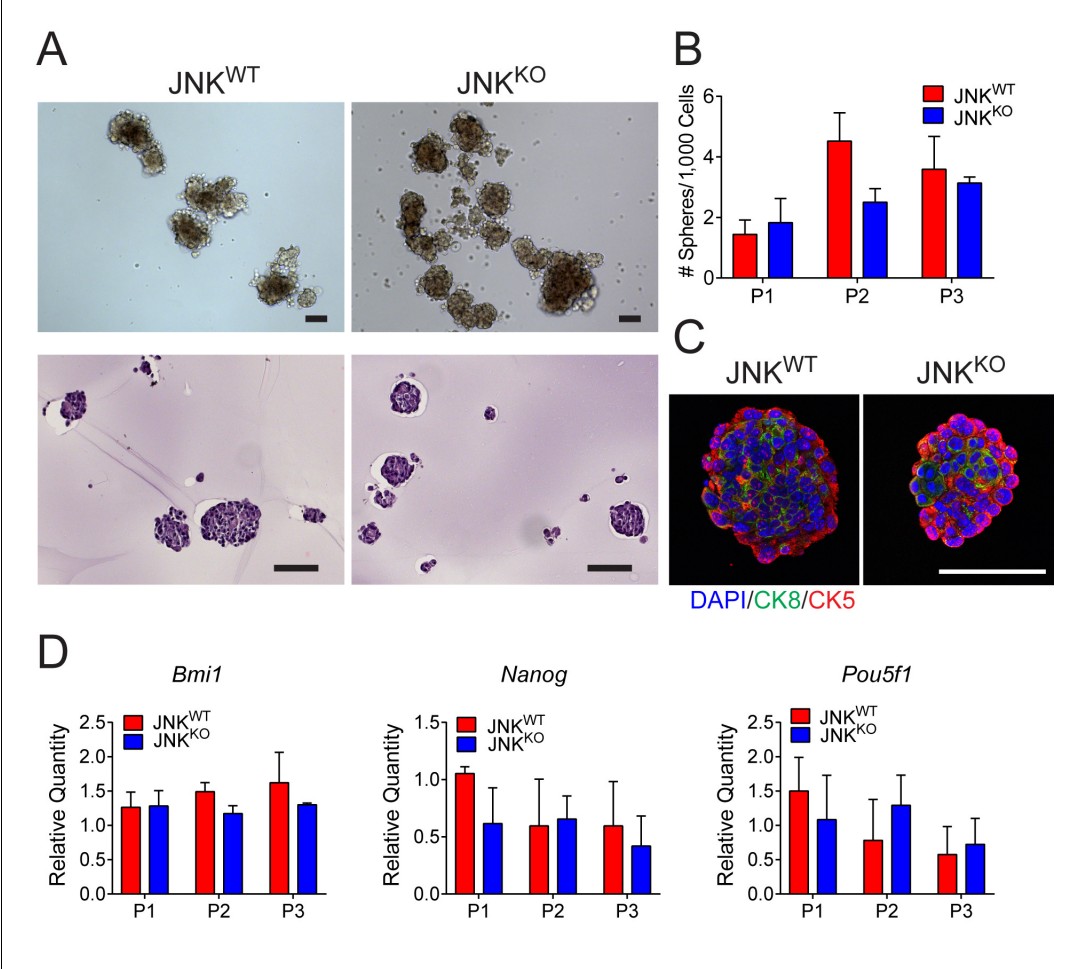

**Figure 5.** Stem cell populations are comparable in JNK^WT and JNK^KO tumor cells. (**A**) JNK^WT and JNK^KO tumor cells formed mammospheres when grown in suspension. Two independent cell lines were tested for each genotype. Representative phase contrast (*upper panel*) and H and E-stained agarose-embedded sphere sections (*lower panel*) are presented. Representative images are presented. Scale bar = 100 µm. (**B**) The number of mammospheres formed per 1000 plated cells each passage (P) was quantitated for JNK^WT and JNK^KO tumor cells. Two independent cell lines were tested for each genotype. The data presented are the mean ± SEM (n = 3 independent experiments). No significant differences were observed (p>0.05). (**C**) Representative agarose-embedded mammosphere sections stained with antibodies to cytokeratin 5 (CK5, red) and cytokeratin 8 (CK8, green), and counterstained with DAPI are presented. Scale bar = 50 µm. (**D**) RNA was isolated from JNK^WT and JNK^KO tumor cell mammospheres at different passages to quantify mRNA expression of *Bmi1*, *Nanog*, and *Pou5f1*. Two independent cell lines were tested for each genotype. The data presented are the mean ± SEM (n = 2 independent experiments). No significant differences were observed (p>0.05).

DOI: https://doi.org/10.7554/eLife.36389.024

The following source data and figure supplements are available for figure 5:

**Source data 1.** Spreadsheet of source data for *Figure 5*.
DOI: https://doi.org/10.7554/eLife.36389.026

**Figure supplement 1.** JNK-deficient tumor cells do not exhibit enhanced tumor stem cell activity.
DOI: https://doi.org/10.7554/eLife.36389.025

**Figure supplement 1—source data 1.** Spreadsheet of source data for *Figure 5—figure supplement 1B*.
DOI: https://doi.org/10.7554/eLife.36389.027

through matrigel-coated membranes (*Figure 6C*) and collagen I-filled wounds (*Figure 6D*) was similar. These data demonstrate that JNK deficiency does not alter breast tumor cell proliferation, invasion, or migration. Consistent with this conclusion, we found no differences between the growth of JNK^WT and JNK^KO tumor cells in orthotopically transplanted syngeneic mice (*Figure 6E–G*).

While many properties of JNK^WT and JNK^KO tumor cells are similar (*Figure 6A–G*), we did detect some JNK-dependent differences in the tumor cell phenotype (*Figure 6H–J*). Thus, JNK^KO tumor

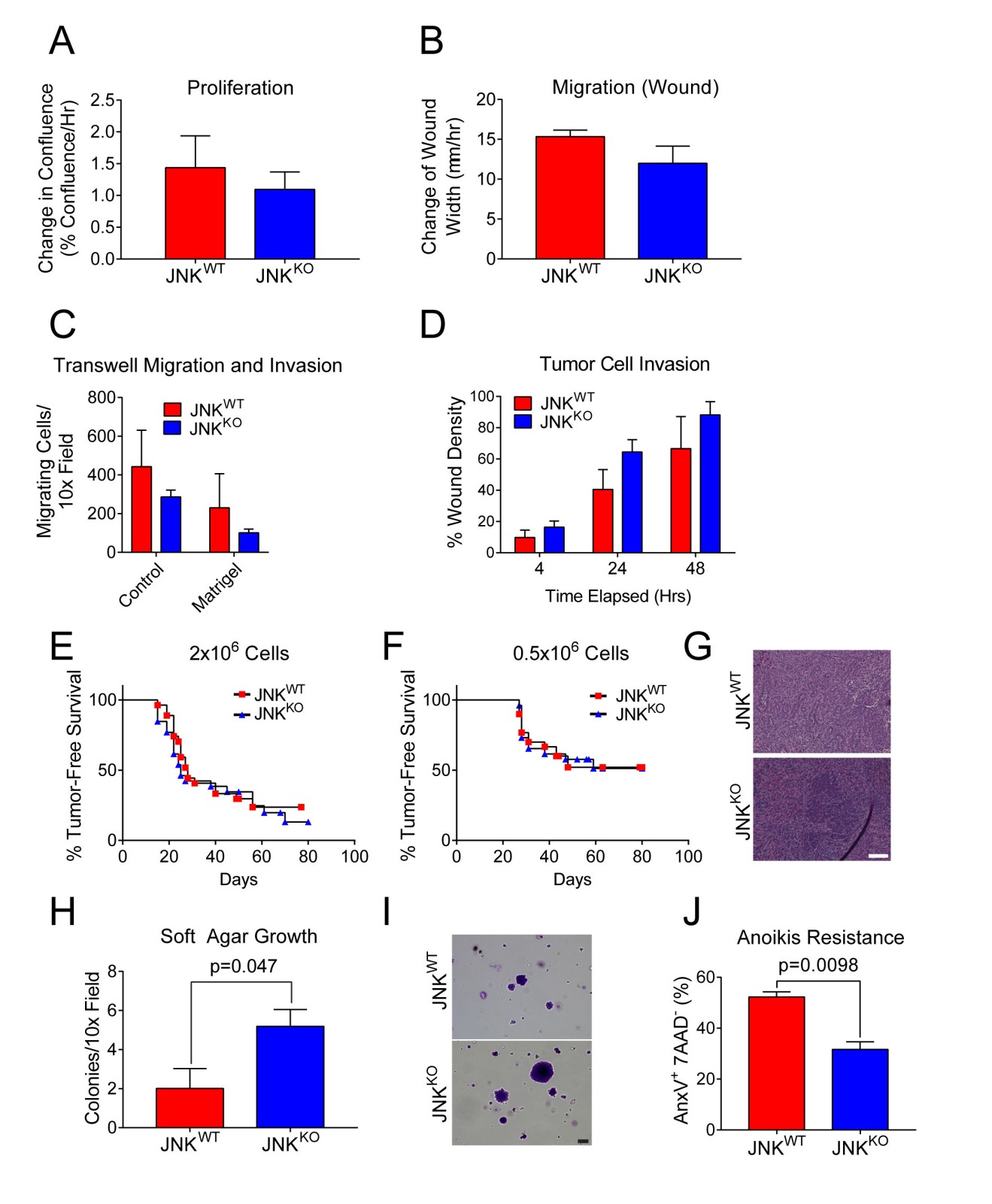

**Figure 6.** JNK[WT] and JNK[KO] tumor cells exhibit similar phenotypes. (A) JNK[WT] (n = 2) and JNK[KO] (n = 6) tumor cell lines were cultured (8 hr) in growth media and the change in confluence was measured using an IncuCyte ZOOM (mean ± SEM). No significant differences (p>0.05) were observed. Similar data were obtained in four independent experiments. (B) Monolayers of JNK[WT] (n = 2) and JNK[KO] (n = 4) cells were wounded and cell migration rates were assessed by measuring the change in wound width 48 hr after wounding using an IncuCyte ZOOM (mean ± SEM). No significant differences

*Figure 6 continued on next page*

*Figure 6 continued*

(p>0.05) were observed. Similar data were obtained in two independent experiments. (**C**) JNK$^{WT}$ (n = 2) and JNK$^{KO}$ (n = 6) tumor cell chemotaxis in response to a serum gradient in the absence (Control) or presence of Matrigel was examined (mean ± SEM). No significant differences were observed (p>0.05). Similar data were obtained in two independent experiments. (**D**) Monolayers of JNK$^{WT}$ (n = 2) and JNK$^{KO}$ (n = 3) tumor cells were wounded, overlayed with 0.5 mg/ml collagen I in growth serum, and cultured for up to 48 hr in media containing 2% serum. Tumor cell migration into the collagen-filled wound was quantitated by measuring cell density in the initial wound area using an IncuCyte ZOOM (mean ± SEM). No significant differences were observed (p>0.05). Similar data were obtained in two independent experiments. (**E–G**) Orthotopic transplantation of JNK$^{WT}$ and JNK$^{KO}$ tumor cells (two independent cell lines per genotype) into the mammary fat pads of 26 (JNK$^{WT}$) and 27 (JNK$^{KO}$) syngeneic wild-type host mice was performed using 2 × 10$^6$ tumor cells (**E**). Orthotopic transplantation of 0.5 × 10$^6$ tumor cells was performed using 30 (JNK$^{WT}$) and 26 (JNK$^{KO}$) syngeneic wild-type host mice (**F**). No significant differences were observed (p>0.05). Representative H and E-stained tumor sections from JNK$^{WT}$ (*upper panel*) and JNK$^{KO}$ (*lower panel*) tumors are presented (**G**). Scale bar = 100 µm. (**H, I**) JNK$^{WT}$ (n = 2) and JNK$^{KO}$ (n = 4) tumor cell lines were cultured in soft agar and colony formation was quantitated (mean ± SEM) (**H**). Similar data were obtained from two independent experiments and representative images of crystal violet-stained colonies are presented (**I**). Scale bar = 100 µm. (**J**) Two JNK$^{WT}$ and five JNK$^{KO}$ tumor cell lines were cultured in suspension (24 hr) and apoptotic cells (7AAD$^-$ annexin V$^+$) were quantitated by flow cytometry (mean ± SEM; n = 7 for JNK$^{WT}$ and n = 16 for JNK$^{KO}$).

DOI: https://doi.org/10.7554/eLife.36389.028

The following source data is available for figure 6:

**Source data 1.** Spreadsheet of source data for *Figure 6*.
DOI: https://doi.org/10.7554/eLife.36389.029

cells formed more colonies than JNK$^{WT}$ tumor cells when grown in soft agar (*Figure 6H,I*) and JNK$^{KO}$ cells exhibited greater resistance to anoikis than JNK$^{WT}$ tumor cells (*Figure 6J*). These data indicate that JNK deficiency can promote tumor cell survival.

## JNK deficiency causes early disease initiation

Comparative studies of JNK$^{WT}$ and JNK$^{KO}$ tumor cells demonstrated that JNK-deficiency does not contribute to differences in proliferation, migration, or invasion phenotypes in vitro (*Figure 6A–D*) or to tumor growth in orthotopically transplanted syngeneic mice (*Figure 6E–G*). We therefore considered the possibility that JNK may influence tumor initiation rather than the function of fully developed tumor cells. To test this hypothesis, we examined sections of mammary glands from female mice at 18 weeks after gene ablation. JNK$^{WT}$ mice presented normal mammary gland morphology (*Figure 7A*). In contrast, multifocal mammary intraepithelial neoplasia (MIN) was observed in JNK$^{KO}$ glands (*Figure 7A*). The presence of MIN lesions in young JNK$^{KO}$ mice indicates that JNK-deficient mammary epithelial cells exhibit defects in apical-basal polarity, a hallmark of mammary tumor development (*Halaoui et al., 2017*; *Zhan et al., 2008*). Moreover, it is likely that mammary tumor development is further promoted by the increased survival of JNK$^{KO}$ cells in vitro (*Figure 6H–J*) and increased proliferation of JNK$^{KO}$ epithelial cells compared with JNK$^{WT}$ epithelial cells in vivo (*Figure 7B,C*). These data indicate that one physiological function of JNK in mammary epithelial cells is to suppress breast cancer development by preventing the initiation of carcinogenesis.

## Discussion

The frequent mutation of the JNK pathway in human breast cancer (including the genes *MAP3K1*, *MAP2K4*, and *MAP2K7*) implicates reduced JNK signaling in the etiology of mammary carcinoma (*Banerji et al., 2012*; *Cancer Genome Atlas Network, 2012*; *Ciriello et al., 2015*; *Ellis et al., 2012*; *Kan et al., 2010*; *Nik-Zainal et al., 2016*; *Shah et al., 2012*; *Stephens et al., 2012*; *Wang et al., 2014*). Mutation of *Mapk8* (encodes JNK1) and *Mapk9* (encodes JNK2) is not frequently detected, most likely because of the functional redundancy of these JNK isoforms (*Davis, 2000*). In contrast, MAP2K4 and MAP2K7 serve as non-redundant activators of JNK and mutation of either *MAP2K4* or *MAP2K7* causes JNK inhibition (*Tournier et al., 2001*). Similarly, MAP3K1 can act in a non-redundant manner to activate JNK (*Yujiri et al., 2000*). Tumor-associated ablation or mutation of these genes therefore causes JNK inhibition. Moreover, JNK signaling is also inhibited by phosphorylation of MAP2K4 by AKT (*Park et al., 2002*) in breast tumors with AKT activation caused by 'driver' mutations in *PTEN* or *PIK3CA* (*Garraway and Lander, 2013*). Suppression of JNK signaling is therefore a characteristic of many breast cancers. Nevertheless, the significance of JNK pathway inactivation in

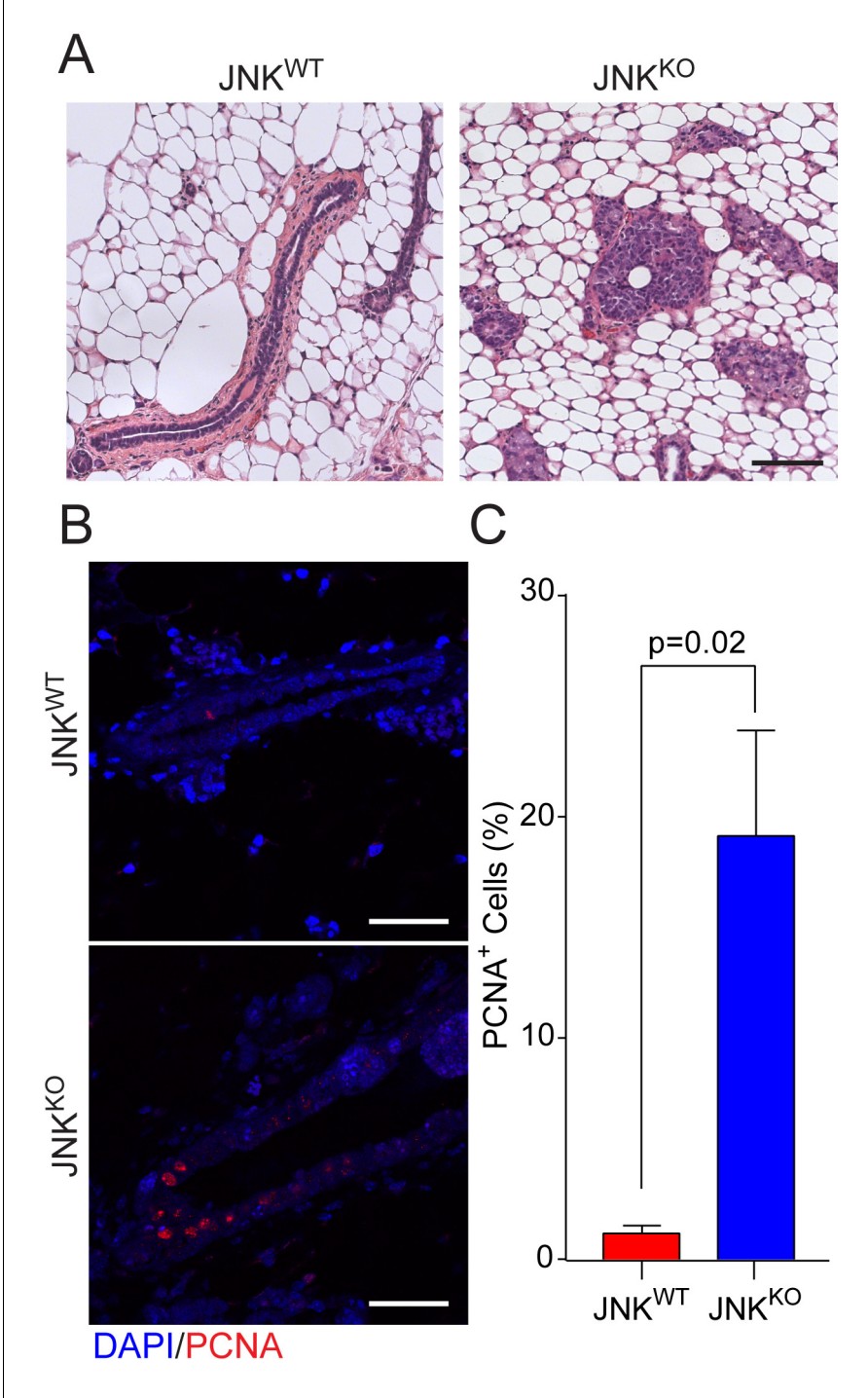

**Figure 7.** JNK deficiency promotes early disease lesions. (**A**) Eighteen weeks after gene deletion, tissue sections were prepared from mammary glands of JNK$^{WT}$ (n = 5) and JNK$^{KO}$ (n = 12) female mice. The mice were not matched for estrus cycle. Representative H and E-stained sections are presented. Scale bar = 100 μm. (**B, C**) Proliferation was examined in mammary glands at 18 weeks after gene ablation by staining tissue sections with an antibody to PCNA (JNK$^{WT}$ n = 4 mice, JNK$^{KO}$ n = 5 mice). Representative PCNA-stained and DAPI counter-stained glands are presented (B, Scale bar = 50 μm). The percent of PCNA$^+$ cells was quantified (mean ± SEM) (**C**).
DOI: https://doi.org/10.7554/eLife.36389.030

The following source data is available for figure 7:

**Source data 1.** Spreadsheet of source data for *Figure 7*.
DOI: https://doi.org/10.7554/eLife.36389.031

breast cancer is unclear (*Cellurale et al., 2012*; *Cellurale et al., 2010*). Here, we demonstrate that loss of JNK signaling promotes murine breast cancer development (*Figure 1*). Furthermore, JNK deficiency accelerated tumor development in a murine model of breast cancer (*Figure 3*). The frequent JNK pathway *loss-of-function* mutations in human breast cancer may therefore represent 'driver' mutations that promote tumor development.

We find that JNK loss plays a key role at the early stages of breast cancer development by promoting mammary gland neoplasia (*Figure 7*). This observation likely accounts for the effect of mutational inactivation of JNK signaling to promote tumor development (*Figure 1 and 3*). In contrast to this tumor-suppressive role of JNK during early stages of carcinogenesis (*Figure 7*), JNK does not markedly influence the late-stage tumor phenotype (*Figure 6*). Thus, JNK does not change tumor cell proliferation, invasion or migration in vitro or tumor formation in orthotopically transplanted mice in vivo (*Figure 6*). However, JNK deficiency does repress tumor cell anoikis and promotes growth in soft agar in vitro (*Figure 6*). JNK therefore plays a major role in early tumor development, but only a minor role in established tumors.

An important question relates to the mechanism of breast tumor suppression by the JNK signaling pathway. Part of the mechanism may reflect the pro-apoptotic role of JNK (*Tournier et al., 2000*). Mutations in key genes can enable epithelial cell survival in the luminal space, resulting in ductal carcinoma in situ (DCIS) and cancer (*Halaoui et al., 2017*; *Leung and Brugge, 2012*; *Pradeep et al., 2012*; *Taraseviciute et al., 2010*). Indeed, JNK deficiency has been shown to impair efficient epithelial cell death in response to cell detachment (anoikis) in vitro and cause both occlusion of mammary ducts (*Cellurale et al., 2012*; *Girnius and Davis, 2017*) and delayed involution (*Girnius et al., 2018*) in vivo. It is therefore possible that these types of cell survival responses in the mammary epithelium may contribute to tumor development. However, this mechanism may not be sufficient to account for the tumor formation caused by JNK deficiency (*Figure 1*) or the observed acceleration of tumor formation caused by JNK deficiency in a mouse model of breast cancer (*Figure 3*). For example, both JNK and p38 MAPK promote mammary gland epithelial cell anoikis by a BIM-dependent mechanism and are required for luminal clearance of mammary ducts (*Cellurale et al., 2012*; *Girnius and Davis, 2017*; *Wen et al., 2011*), but JNK inhibition promotes breast cancer (*Figure 1 and 3*) while p38 MAPK pathway inhibition suppresses breast cancer development and promotes the early dissemination of tumor cells (*Del Barco Barrantes et al., 2018*; *Gawrzak et al., 2018*; *Harper et al., 2016*). These data indicate that anoikis defects and ductal occlusion by detached epithelial cells caused by stress-activated MAPK inhibition is not sufficient for tumor formation. This reasoning implicates a second pathway of JNK-mediated suppression of mammary tumor development.

Insight into JNK-mediated tumor suppression was obtained from the analysis of tumor genomic DNA. We found that JNK deficiency caused genomic instability associated with widespread SNVs, Indels, and CNVs (*Figure 1G*). It is likely that these genetic changes caused by JNK deficiency contribute to tumor development. This genetic instability reflects the function of JNK to promote genome maintenance in response to stress. Examples include JNK-mediated phosphorylation of SIRT6 to stimulate double-stranded DNA break repair (*Van Meter et al., 2016*) and JNK-mediated phosphorylation of DGCR8 to induce transcription-coupled nucleotide excision repair (*Calses et al., 2017*). Loss of JNK signaling would therefore be expected to cause defects in DNA repair that result in increased genomic instability. Indeed, our analysis demonstrates that JNK deficiency in the mammary epithelium causes genomic instability (*Figure 1G*) and breast cancer (*Figure 1C*). JNK deficiency may therefore increase the sensitivity of tumor cells to drugs that cause DNA damage. This represents a potential opportunity for the design of therapeutic strategies for the treatment of tumors with defects in JNK signaling.

The TRP53 pathway is known to be important for genome stability and tumor suppression. Our data demonstrate that the JNK pathway similarly promotes genome stability and tumor suppression. The number of SNVs and indels caused by TRP53 deficiency was greater than that caused by JNK deficiency. Nevertheless, the number of CNVs were similar on both genetic backgrounds. These observations indicate that while both TRP53-deficiency and JNK-deficiency can cause genomic instability, the mechanisms engaged by these two pathways may be different.

The identification of JNK pathway mutations as 'driver' mutations for breast cancer was not anticipated. For example, the JNK pathway can promote cell proliferation and survival through regulation of AP1 transcription factors (*Das et al., 2007*; *Lamb et al., 2003*; *Ventura et al., 2003*). Indeed, the

JNK/JUN signaling pathway may promote the proliferation of some breast tumor cells in vitro (*Guo et al., 2006*; *Xie et al., 2017*). It is therefore possible that JNK signaling may have pro-oncogenic functions in certain tumor types or under specific conditions (*Whitmarsh and Davis, 2007*). Nevertheless, our analysis establishes that a major role of JNK signaling in the breast epithelium is tumor suppression (*Figure 1 and 3*). This role of JNK signaling is consistent with the discovery of frequent JNK pathway gene ablation and mutation in human breast cancer (*Banerji et al., 2012*; *Cancer Genome Atlas Network, 2012*; *Ciriello et al., 2015*; *Ellis et al., 2012*; *Kan et al., 2010*; *Nik-Zainal et al., 2016*; *Shah et al., 2012*; *Stephens et al., 2012*; *Wang et al., 2014*).

The major conclusions of this study are that: (1) JNK deficiency in the mammary epithelium can cause murine breast cancer; and (2) JNK deficiency rapidly accelerates tumor development in a mouse model of breast cancer. These observations indicate that JNK can act as a breast cancer tumor suppressor. This is a significant finding because frequent mutational inactivation of genes that encode JNK pathway components (e.g. *MAP2K4* and *MAP3K1*) are detected in human breast cancer (*Garraway and Lander, 2013*). However, it should be noted that the compound disruption of JNK expression that we have studied most likely differs from the loss of individual JNK pathway components (e.g. MAP2K4 and MAP3K1) because of partial compensation by alternative signaling molecules, including MAP2K7 and members of the MAP3K family. Whether reduced JNK activity caused by loss of MAP2K4 or MAP3K1 phenocopies the effects of compound JNK deficiency on breast cancer is unclear. An important goal for future studies will be to directly test the effect of *MAP2K4* and *MAP3K1* gene mutation on breast cancer.

## Materials and methods

### Animals

We have previously described $Mapk8^{LoxP/LoxP}$ $Mapk9^{LoxP/LoxP}$ mice (*Han et al., 2013*). C57BL6/J (RRID:IMSR_JAX:000664), B6129-TG(Wap-cre)11738Mam/J (RRID:IMSR_JAX:008735) (*Wagner et al., 1997*), B6.129P2-$Trp53^{tm1Brn}$/J (RRID:IMSR_JAX:008462) (*Marino et al., 2000*), and B6.129(Cg)-Gt(ROSA)26Sor$^{tm4(ACTB-tdTomato,-EGFP)Luo}$/J mice (RRID:IMSR_JAX:007676) (*Muzumdar et al., 2007*) were purchased from The Jackson Laboratories (Bar Harbor, ME). All mice were on a C57BL6/J strain background (10 generations back-crossed). Female mice were bred at 10–12 weeks of age and monitored for tumor development by palpation following weaning. Reasons for euthanasia included a tumor of 1 cm in diameter, ulceration due to the tumor, poor health, or weight loss (>20% body weight). The animals were housed in a specific pathogen-free facility accredited by the American Association of Laboratory Animal Care (AALAC).

### Tumor-derived cell lines

Tumors were harvested and placed in DMEM/F12 medium supplemented with 1% penicillin/streptomycin and 1% nystatin (Thermo Fisher Scientific, Waltham, MA). The tissue was washed with phosphate-buffered saline (PBS), minced, and digested (37°C for up to 2 hr, shaking) in DMEM supplemented with 2 mg/ml collagenase, 0.1% trypsin, and 1% penicillin/streptomycin. Following digestion, cells were pelleted and re-suspended in DMEM/F12 supplemented with 2% fetal bovine serum and 1% penicillin/streptomycin, then plated on collagen I (Thermo Fisher Scientific Cat# A1048301) coated dishes. After two passages, the cells were plated on standard tissue culture dishes and maintained in Growth Medium (DMEM/F12 supplemented with 10% fetal bovine serum plus 1% penicillin/streptomycin) prior to cryogenic storage.

### Genotype analysis

Genomic DNA was genotyped using a PCR-based method. $Mapk8^{LoxP}$ (540 bp) and $Mapk8^+$ (330 bp) were detected using amplimers 5'-AGGATTTATGCCCTCTGCTTGTC-3' and 5'-GACCACTGTTCCAATTTCCATCC-3'. $Mapk9^{LoxP}$ (264 bp) and $Mapk9^+$ (224 bp) were detected using amplimers 5'-GTTTTGTAAAGGGAGCCGAC-3' and 5'-CCTGACTACTGAGCCTGGTTTCTC-3'. $Trp53^{LoxP}$ (370 bp) and $Trp53^+$ (288 bp) were detected using amplimers 5'-AGCACATAGGAGGCAGAGAC-3' and 5'-CACAAAAACAGGTTAAACCCAG-3'. $Cre^+$ (450 bp) was detected using amplimers 5'-TTACTGACCGTACACCAAATTTGCCTGC-3' and 5'-CCTGGCAGCGATCGCTATTTTCCATGAGTG-3'. $Mapk8^{\Delta}$ (395 bp), $Mapk8^+$ (1550 bp), and $Mapk8^{LoxP}$ (1,095 bp) were detected using amplimers 5'-

CCTCAGGAAGAAAGGGCTTATTTC-3' and 5'-GAACCACTGTTCCAATTTCCATCC-3'. *Mapk9*$^\Delta$ (400 bp) and *Mapk9*$^{LoxP}$ (560 bp) were detected using 5'-GGAATGTTTGGTCCTTTAG-3', 5'-GCTATTCA-GAGTTAAGTG-3', and 5'-TTCATTCTAAGCTCAGACTC-3'. *Trp53*$^\Delta$ (612 bp) was detected using amplimers 5'-GAAGACAGAAAAGGGGAGGG-3' and 5'-CACAAAAACAGGTTAAACCCAG-3'.

## Orthotopic transplantation

Tumor-derived cell lines were screened for common pathogens using the Rapidmap panel 21 (Taconic Biosciences, New York, NY). The cells were pelleted and washed five times with PBS before resuspension at a final concentration of $2 \times 10^6$ or $0.5 \times 10^6$ cells/40 µl of PBS. A 26-guage needle was used to inject the cells into the thoracic mammary gland. The presence of tumors was monitored weekly and palpable tumors were measured using calipers. Mice meeting euthanasia criteria or mice that remained 80 days post-inoculation were euthanized.

## Cell proliferation

Cells were seeded in 96-well plates (1,000 cells/well) and cultured in an IncuCyte Zoom (Essen Bioscience). The cells were kept in growth media and fed every 2 days for the duration of the assay. Images were taken every 4 hr and cell confluence was measured at these time points.

## Soft agar growth

Agarose (Lonza, Basel, CH) was dissolved in water (4 or 2% (w/v) solution) and autoclaved. The 4% agarose was mixed with growth media to yield a 1% agarose solution, which was plated and allowed to solidify (4°C). Tumor cells in growth media (100,000/6 cm dish) were mixed with the 2% agarose to a final concentration of 0.5% agarose, and overlayed onto the solidified 1% agarose. The cells were fed with Growth Medium every 3 ~ 4 days for the duration of the assay. After 3 weeks, the cells were fixed with 100% methanol (−20°C) and stained with 0.1% crystal violet dissolved in 20% methanol/80% PBS. Images were taken using a Zeiss AxioVert 200M microscope. Six 10x-fields/plate were used to quantify colony formation ($\geq$3000 pixels$^2$) using FIJI software (*Schindelin et al., 2012*).

## Mammosphere assay

Tumor cells were seeded 25,000/ml in a 24-well ultra-low attachment plate in DMEM/F12 media containing 1% penicillin/streptomycin, B27 supplement (1:50), 20 ng/ml bFGF, 2.5 µg/ml insulin, and 20 ng/ml EGF. Six wells were plated for each cell line. Three to four days later, before passaging, colonies larger than 50 µm were counted using a Zeiss AxioVert 200M microscope. The spheres were dissociated by incubation (15 min) with 0.25% trypsin (Thermo Fisher Scientific) with pipetting. The digestion was terminated by addition of 0.5% soybean trypsin inhibitor (ATCC) and single cells were plated. Spheres used for embedding in agarose or for RNA isolation were treated similarly, but were cultured in 100 mm ultra-low attachment dishes. Spheres prepared for embedding were resuspended in 4% agarose (IBI Scientific Cat# IB70050) and fixed in 10% formalin before processing.

## Anoikis assay

Tumor cells were suspended in serum-free DMEM/F12 media with 0.5% methylcellulose (Millipore-Sigma), placed in poly-HEMA (Millipore-Sigma) coated plates ($1.2 \times 10^5$ cells/ml), and incubated for 24 hr. After incubation, the cells were washed twice with PBS, then stained with phycoerythrin-conjugated Annexin V and 7-aminoactinomycin D (7-AAD) using the PE annexin apoptosis detection kit I (BD Pharmingen #559763). A FACSCalibur (BD Bioscience) was used to quantify the apoptotic cells (7AAD$^-$ Annexin V$^+$). Single-stained controls were used to gate 7AAD$^-$ and 7AAD$^+$ cells while cells suspended for 1 hr were used to define the annexin V$^+$ and annexin V$^-$ populations. FlowJo version 9.7.6 (Tree Star) was used to analyze the data (*Girnius and Davis, 2017*).

## Wound healing

Tumor cells were seeded at a density of 100,000 cells/well in a 96-well ImageLock plate (Essen Bioscience) and allowed to adhere overnight. The following day, a Wound Maker (Essen Bioscience) was used to scratch the confluent monolayers of cells. The cells were washed two times with PBS and then maintained in serum-free media for the remainder of the assay. Invasion assays were performed using cells plated in 96-well ImageLock plates coated with 300 µg/ml collagen I. Cells

adhered overnight, were scratched using a WoundMaker, and washed twice with media before being placed on ice (5 min). The cells were then overlayed with growth media containing collagen I (0.5 mg/ml). The cells were incubated (30 min at 37°C) to solidify the collagen prior to the addition of DMEM/F12 supplemented with 2% FBS. All plates were imaged every 4 hr using an IncuCyte Zoom (Essen Bioscience).

## Transwell migration

Tumor cells (30,000 cells/well) were plated in triplicate using Growth Medium in transwells with 8 μm pores (Millipore Sigma) and allowed to adhere. For invasion assays, inserts coated with Matrigel (Millipore Sigma) were rehydrated according to the manufacturer's instructions before plating cells. To create a serum gradient, media from the upper chambers was replaced with serum-free media. Cells were allowed to migrate for 16–20 hr. Remaining cells in the upper chambers were removed with a cotton swab, then the inserts were fixed with 100% methanol (20 min, −20°C), washed three times with PBS, and stained with 2-(4-amidinophenyl)−1 hr -indole-6-carboxamidine (DAPI). The insert membranes were removed with a scalpel and mounted on a slide for examination using a Zeiss Axiovert 200M microscope. Two to three images at 10x magnification were taken for each membrane. FIJI was used to quantify the migrating cells (*Schindelin et al., 2012*).

## Histological analysis

Mammary glands #2–5 were harvested, fixed in 10% formalin, dehydrated, and embedded in paraffin. Hematoxylin and eosin-stained sections (5 μm) were reviewed by a board-certified veterinary pathologist to identify and classify proliferative lesions (*Cardiff et al., 2000*). To detect proliferation, sections were treated with the endogenous biotin blocking kit (Thermo Fisher Scientific E21390), prior to incubation with a biotin-conjugated antibody to PCNA (Thermo Fisher Scientific Cat# 13–3940 RRID:AB_2533017; dilution 1:50), and an AlexaFluor 633-conjugated streptavidin (Thermo Fisher Scientific Cat# S-21375 RRID:AB_2313500). For PCNA quantification 49–50 fields were examined across JNK$^{WT}$ (n = 4) and JNK$^{KO}$ (n = 5) mice. PCNA positive duct cells were normalized to the total number of duct cells in the field examined. Additional sections were stained with cleaved-caspase-3 (Cell Signaling Technology Cat# 9662 RRID:AB_331439; 1:100), cytokeratin 5 (BioLegend Cat# 905501 RRID:AB_2565050; 1:50), cytokeratin 8 (DSHB Cat# TROMA-I RRID:AB_531826; 1:100), GFP (Thermo Fisher Scientific Cat# A21311 RRID:AB_221477; 1:100), estrogen receptor (Santa Cruz Biotechnology Cat# sc-542 RRID:AB_631470; 1:500), and progesterone receptor (Santa Cruz Biotechnology Cat# sc-538 RRID:AB_632263; 1:300). For immunohistochemistry, a biotinylated rabbit antibody (Biogenex Cat# HK340-5K) in conjunction with streptavidin-conjugated horseradish peroxidase (Vector Laboratories Cat# PK-6100) and 3,3'-diaminobenzidene (Vector Laboratories Cat# SK-4100) was used to detect the primary antibody. Sections were then counterstained with hematoxylin (Fisher Scientific) and pictures were taken using a Zeiss Axiovert. AlexaFluor 546 conjugated-goat anti-rabbit IgG (H + L) antibody (Thermo Fisher Scientific Cat# A11035 RRID:AB_143051) and AlexaFluor 488 conjugated-goat anti-rat IgG (H + L) antibody (Thermo Fisher Scientific Cat# A11006 RRID:AB_141373) were used to detect immune complexes in co-staining experiments. These sections were counterstained with DAPI and immunofluorescence was examined using a Leica SP2 confocal microscope.

## Immunoblot analysis

Cell lysates were prepared using Triton lysis buffer (20 mM Tris [pH 7.4], 1% Triton X-100, 10% glycerol, 137 mM NaCl, 2 mM EDTA, 25 mM β-glycerophosphate, 1 mM sodium orthovanadate, 1 mM phenylmethylsulfonyl fluoride, and 10 μg/mL of aprotinin and leupeptin). Extracts (30 μg) were examined by immunoblot analysis by probing with antibodies to phospho-ERK (Cell Signaling Technology Cat# 9101 RRID:AB_331646), ERK2 (Santa Cruz Biotechnology Cat# sc-1647 RRID:AB_627547), phospho-JNK (Cell Signaling Technology Cat# 9255 RRID:AB_2307321), JNK (R and D Systems Cat# AF1387 RRID:AB_2140743), p38 (Cell Signaling Technology Cat# 9212 RRID:AB_330713), phospho-p38 (Cell Signaling Technology Cat# 9211 also 9211L, 9211S RRID:AB_331641), phospho-JUN (S63) (Cell Signaling Technology Cat# 9261L RRID:AB_2130159), JUN (Santa Cruz Biotechnology Cat# sc-1694 RRID:AB_631263), p-AKT (T308) (Cell Signaling Technology Cat# 5106S RRID:AB_836861), p-AKT (S473) (Cell Signaling Technology Cat# 9271 RRID:AB_329825), AKT (Cell Signaling

Technology Cat# 9272 RRID:AB_329827), and α-Tubulin (Millipore-Sigma Cat# T5168 RRID:AB_477579). IRDye 680LT conjugated-donkey anti-mouse IgG antibody (LI-COR Biosciences Cat# 926–68022 RRID:AB_10715072) and IRDye 800CW conjugated-goat anti-rabbit IgG (LI-COR Biosciences Cat# 926–32211 RRID:AB_621843) were used to detect and quantitate immune complexes with the Odyssey infrared imaging system (LI-COR Biosciences).

## Exome sequencing

Genomic DNA was isolated from cancer cell lines using the DNEasy kit, including RNase treatment (Qiagen). For each genotype, a control sample was generated by isolating genomic DNA from mammary glands of a virgin mouse of the same genotype. Whole exome sequencing libraries were prepared using the Agilent SureSelect XT library preparation kit. DNA (OD260/280 1.7–2.0 and OD260/230 > 2.0) was sheared using a Covaris LE220. End-repaired, adenylated DNA fragments were ligated to Illumina sequencing adapters and amplified by PCR. Exome capture was performed using the Agilent SureSelect Mouse All Exon (50 mb) capture probe set; captured exome libraries were enriched by PCR. Final libraries were quantified using the KAPA Library Quantification Kit (KAPA Biosystems), Qubit Fluorometer (Life Technologies) and Agilent 2100 BioAnalyzer, and were sequenced on an Illumina HiSeq2500 machine using $2 \times 125$ bp cycles. Single nucleotide variants were reported as the union of SNVs called by muTect (*Cibulskis et al., 2013*), Strelka (*Saunders et al., 2012*), and LoFreq (*Wilm et al., 2012*) and indels were reported as the union of indels called by Strelka, somatic versions of Pindel (*Ye et al., 2009*) and Scalpel (*Narzisi et al., 2014*). ExomeCNV was run with default settings using mm10 reference to generate copy-number calls (*Sathirapongsasuti et al., 2011*). The segmentation and $\log_2$ ratios from ExomeCNV output were used to identify amplified, deleted and copy-neutral regions. $\log_2$ thresholds of $>0.2$ and $<-0.2$ were used to label a segment as amplified or deleted, respectively, and the segments were visualized (*Krzywinski et al., 2009*). Bedtools (*Quinlan and Hall, 2010*) was run to identify genes overlapping copy-number segments.

## Analysis of mRNA expression

Cellular RNA was isolated using the RNeasy kit with DNase treatment (Qiagen) and RNA quality (RIN >9) was confirmed using the Bioanalyzer 2100 system (Agilent Technologies). Libraries were constructed according to the manufacturer's instructions (Illumina). Two to three libraries were analyzed for each condition. Single-end sequencing with reads of 40 bp reads (for JNK$^{WT}$ and JNK$^{KO}$) was performed on an Illumina HiSeq 2000 platform and paired-end sequencing with 150 bp reads (for MEC and ME$^{KO}$) was performed on an Illumina NextSeq500 platform (*Figure 2*). Poor quality reads, adapter sequences, and reads less than 40 bp were removed using Trimmomatic (version 0.36) (*Bolger et al., 2014*). First, the MEC and ME$^{KO}$ biological groups were analyzed (*Figure 2*). Second, to adequately compare the four biological groups (MEC, ME$^{KO}$, JNK$^{WT}$, and JNK$^{KO}$), seqtk was used to sample sequences averaging 40 million single-end reads per sample (*Figure 4—figure supplement 1A*) (*Li, 2017*). Reads were aligned to the mouse genome mm10 using Bowtie2 (v 2–2.1.0) (*Langmead and Salzberg, 2012*) and Tophat2 (v 2.0.14) (*Kim et al., 2013*). Samtools (version 0.0.19) (*Li et al., 2009*) and IGV (version 2.3.60) (*Thorvaldsdóttir et al., 2013*) were used for indexing the alignment files and viewing the aligned reads respectively. Cufflinks (v 2.2.1) (*Trapnell et al., 2012*; *Trapnell et al., 2010*) was used to quantitate gene expression as fragments per kilobase of exon model per million mapped fragments (FPKM); differential expression was identified using the Cufflinks tools, Cuffmerge and Cuffdiff. Cummerbund version 2.4.1 (*Trapnell et al., 2012*) was used to assess replicate concordance. The complex heatmap package version 1.12.0 (*Gu et al., 2016*) was used to generate heatmaps of differentially expressed (q < 0.05, |$\log_2$ Fold Change| > 0.75) genes.

Pathway over-representation analysis was performed on differentially expressed genes using the WEB-based GEne SeT AnaLysis Toolkit (Webgestalt) (*Wang et al., 2013*) with the KEGG database. This analysis was also performed with gene set enrichment analysis (GSEA) software (version 3.0) (*Subramanian et al., 2005*) using ranked genes (sign of the $\log_2$ fold change times the $\log_{10}$ p-value) (*Plaisier et al., 2010*) and the MSigDB gene sets: HALLMARK_KRAS_SIGNALING_UP (M5953); and HALLMARK_DNA_REPAIR (M5898). The gene symbols for the mouse equivalents were used in the gene set enrichment analysis using Ensembl BioMart v87 (*Smedley et al., 2015*).

Ingenuity Pathway Analysis (Qiagen) was used to predict pathway activation based on differentially expressed genes. The top 100 pathways ranked by score (the sum across tumor genotypes of -$\log_{10}$ p-value calculated by Fisher's exact test) were identified. Treeview (Java) (*Saldanha, 2004*) was used to make heatmaps showing pathway activation z-scores.

The expression of mRNA was also determined by RT-PCR using a Quantstudio 12K Flex machine (Thermo Fisher Scientific). TaqMan assays were used to quantify the expression of *Bmi1* (Mm03053308_g1), *Pou5f1* (Mm03053917_g1), and *Nanog* (Mm02019550_s1). Relative expression was normalized to the expression of *18S* RNA in each sample using Taqman assays (catalog number 4308329; Thermo Fisher Scientific).

## Statistical analysis

Data are presented as the mean and standard error. Statistical analysis was performed using Graph-Pad Prism version 7 (GraphPad Software). ANOVA with Bonferroni's test was used to determine significance with an assumed confidence interval of 95%. Two-tailed, unpaired t-test with Welch's correction was used for pairwise comparisons. Fisher's Exact Test was used to determine differences in tumor or MIN incidence. Kaplan-Meier analysis of mammary tumor-free survival was performed using the log-rank test. Statistical significance was defined as $p < 0.05$.

## Data availability

Raw data are presented in the Source Data file. Flow cytometry data have been deposited with FlowRepository (Repository ID:FR-FCM-ZYEV). The RNA-seq data have been deposited with NCBI; GEO accession numbers GSE100581 and GSE92560. The exome sequence data were deposited with NCBI; SRA accession number SRP117075.

## Acknowledgements

We thank the New York Genome Center and the MIT BioMicro Center for assistance with sequence analysis, Armanda Roy for expert technical assistance, and Kathy Gemme for administrative assistance. RJD is an investigator of the Howard Hughes Medical Institute. The authors declare no competing financial interests.

## Additional information

### Competing interests

Roger J Davis: Reviewing editor, *eLife*. The other authors declare that no competing interests exist.

### Funding

| Funder | Grant reference number | Author |
| --- | --- | --- |
| Howard Hughes Medical Institute | Investigator | Roger J Davis |
| National Institute of Diabetes and Digestive and Kidney Diseases | DK107220 | Roger J Davis |
| National Institute of Diabetes and Digestive and Kidney Diseases | DK112698 | Roger J Davis |

The funders had no role in study design, data collection and interpretation, or the decision to submit the work for publication.

### Author contributions

Nomeda Girnius, Conceptualization, Data curation, Formal analysis, Investigation, Writing—original draft, Writing—review and editing; Yvonne JK Edwards, Data curation, Formal analysis, Writing—

review and editing; David S Garlick, Formal analysis, Writing—review and editing; Roger J Davis, Conceptualization, Formal analysis, Supervision, Writing—original draft, Writing—review and editing

### Author ORCIDs
Nomeda Girnius (iD) https://orcid.org/0000-0002-7267-5950
Roger J Davis (iD) https://orcid.org/0000-0002-0130-1652

### Ethics

Animal experimentation: This study was performed in strict accordance with the recommendations in the Guide for the Care and Use of Laboratory Animals of the National Institutes of Health. All of the animals were handled according to approved institutional animal care and use committee (IACUC) protocols (#A-1032) of the University of Massachusetts Medical School.

### Decision letter and Author response
Decision letter https://doi.org/10.7554/eLife.36389.043
Author response https://doi.org/10.7554/eLife.36389.044

## Additional files

### Supplementary files
• Transparent reporting form
DOI: https://doi.org/10.7554/eLife.36389.032

### Data availability

Sequencing data have been deposited with NCBI. Flow cytometry data have been deposited with Flow Repository. All other data are provided as source data files published with this manuscript.

The following datasets were generated:

| Author(s) | Year | Dataset title | Dataset URL | Database, license, and accessibility information |
|---|---|---|---|---|
| Girnius, Edwards, Garlick, Davis | 2018 | The cJun NH2-terminal kinase (JNK) signaling pathway promotes genome stability and prevents tumor initiation | http://www.ncbi.nlm.nih.gov/geo/query/acc.cgi?acc=GSE100581 | Publicly available at the NCBI Gene Expression Omnibus (accession no. GSE100581) |
| Girnius, Edwards, Garlick, Davis | 2018 | The cJun NH2-terminal kinase (JNK) signaling pathway promotes genome stability and prevents tumor initiation | http://www.ncbi.nlm.nih.gov/geo/query/acc.cgi?acc=GSE92560 | Publicly available at the NCBI Gene Expression Omnibus (accession no. GSE92560) |
| Girnius, Edwards, Garlick, Davis | 2018 | The cJun NH2-terminal kinase (JNK) signaling pathway promotes genome stability and prevents tumor initiation | http://trace.ncbi.nlm.nih.gov/Traces/sra/sra.cgi?study=SRP117075 | Publicly available at the NCBI Sequence Read Archive (accession no. SRP117075) |
| Girnius, Edwards, Garlick, Davis | 2018 | The cJun NH2-terminal kinase (JNK) signaling pathway promotes genome stability and prevents tumor initiation | http://flowrepository.org/id/FR-FCM-ZYEV | Publicly available at FlowRepository (accession no. FR-FCM-ZYEV) |

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
