## [Decision Letter]

Thank you for submitting your article "The cJUN NH_2_-terminal kinase (*JNK*) signaling pathway promotes genome stability and prevents tumor initiation" for consideration by *eLife*. Your article has been reviewed by three peer reviewers, and the evaluation has been overseen by a Reviewing Editor and Jonathan Cooper as the Senior Editor. The reviewers have opted to remain anonymous.

The reviewers have discussed the reviews with one another and the Reviewing Editor has drafted this decision to help you prepare a revised submission.

Your paper has been seen by three reviewers, and myself, and we are very happy to consider publication after some revisions that potentially include some new data, some points of clarification/further discussion, and some addressing of minor points. Overall, all three reviewers consider the work presented to be novel and interesting, adding significantly to the existing literature on the role of *JNK* in tumorigenesis, and the context dependency of that. There are some suggested additional data that would enhance mechanistic advance if these can be added, and quite a few points of clarification or discussion that the reviewers felt would be important and/or add to the paper. Some of these were considered important to avoid slightly superficial interpretation of some data in places, particularly around what the model used here was actually reporting. These points are outlined below. We look forward to seeing a revised version.

*Major points:*

1) The authors cannot use the wording that *JNK* deletion is 'sufficient' to cause tumors? Only 10 of 32 ME-ko mice had tumors. For Figure 1C the authors need to be explicit in the ages of the mice under observation. What is the latency period of the tumors and MIN lesions? How long were control animals followed? It is surprising the authors have chosen JNKf/f (Cre-negative) animals as their control. Was any *WAP-Cre;JNK*-wt cohort aged during the study?

2) The main conclusion is that *JNK* suppresses breast cancer initiation; however the authors have recently published that *JNK* deletion results in a delay in mammary gland involution (Girnius et al., CDD). Therefore, could the lesions observed in Figure 7 be due to a lack of clearance post-weaning? This could lead to an expanded pool of cells susceptible to transformation and so a quirk of the *WAP-Cre* model. Have the authors considered using an alternative mammary-specific Cre driver such as *K8-Cre* or *MMTV-Cre* which would obviate the need to induce lactation? At the very least, this should be discussed in more detail. Can *JNK* be deleted in established tumors, either using an inhibitor or perhaps by way of an inducible knockdown orthotopic model?

3) There are 3 examples of ME-KO tumours in Figure 1D but the authors need to say if these are adenocarcinoma or adenosquamous carcinoma. Ideally at least two examples of each tumor type should be shown. It is not clear if the adenosquamous carcinomas originate in a *JNK*-deleted cell type since the *WAP-cre* driver is CK8-specific (Figure 1A) yet the adenosquamous lesions are predominately CK5 positive. Furthermore, there is no evidence for the trans-differentiation unless the authors can show these CK5+ tumours are *JNK*-deleted.

4) The RNAseq data presented in Figure 2 demonstrates an upregulation of WNT target genes in ME^KO^ cells, leading the authors to suggest that ME^KO^ tumors could exploit WNT signalling to drive proliferation. Validating this observation at the protein level through IHC of tumor tissue or Western blotting of ME^WT^ and ME^KO^ cell lines would add confidence to this proposed mechanism. Are WNT pathway components differentially expressed between *JNK*^WT^ and *JNK*^KO^ tumor cells in the p53mut background? Based on data in Figure 2E and 2F, would authors predict that *JNK*^KO^ tumor cells should be more sensitive to WNT pathway inhibitors? Has this been tested?

5) The detection of genomic instability indicators in ME^KO^ tumor cell lines is interesting but perhaps expected when comparing tumor cells with normal epithelial cells. It would have been useful to compare ME^KO^ with ME^WT^ tumor cell lines. It does looks like in the p53mut background, *JNK*^KO^ cells have less SNVs and indels than *JNK*^WT^ cells (Figure 1—figure supplement 2A). Authors should comment on this.

6) Do gene expression changes associated with *JNK* deficiency in mammary tumor cells correlate with previously published gene signatures associated with genomic instability or DNA damage?

*Points for clarification/discussion:*

7) Human breast cancers retain expression of JNK1 and JNK2 but exhibit inactivating mutations in upstream components, which will strongly impair activation of *JNK*. This is obviously not the same as the complete ablation of *JNK* expression that the authors employ in this model. Whilst the data presented are strong and provide important new insights into the role of *JNK* in this murine model, the authors do need to explicitly acknowledge and comment on this limitation of their model.

8) At several places in the manuscript (Introduction and Conclusion) the authors state that their discovery that loss of JNK1/2 expression promotes breast cancer may 'present an opportunity for therapeutic intervention'. It is not immediately obvious how a small molecule could rescue the effects of losing JNK1/2 expression to prevent genome instability. Presumably they are thinking about a disease stratification approach in which those breast cancers with loss of function mutations on the *JNK* pathway and enhanced genome stability may be more susceptible to intervention with drugs targeting DNA Damage Response (DDR) pathways, analogous to synthetic lethality that has helped to drive approval of PARPi? Or do they envisage combination of JNKi and DDRi as being synthetic lethal in cells that do not have *JNK* pathway mutations? A little more information here, or discussion, would help to clarify ideas and potential significance for the reader.

9) In Figure 3 and Figure 3—figure supplement 2 the authors describe how the *Kras* locus is more frequently amplified in JNK^WT^ mice compared to JNK^KO^. Could the authors comment on the significance of this observation? Are they proposing that because the JNK^KO^ tumours display increased genome instability they are likely to acquire many more oncogenic events and thus become less reliant on the amplification of the *Kras* locus which is prevalent in the JNK^WT^ mice?

---

## [Author Response]

Major points:

1) The authors cannot use the wording that JNK deletion is 'sufficient' to cause tumors? Only 10 of 32 ME-ko mice had tumors. For Figure 1C the authors need to be explicit in the ages of the mice under observation. What is the latency period of the tumors and MIN lesions? How long were control animals followed? It is surprising the authors have chosen JNKf/f (Cre-negative) animals as their control. Was any WAP-Cre;JNK-wt cohort aged during the study?

a) We agree – we were wrong to state that *JNK*-deficiency is sufficient for tumor formation since tumors were not found in all *JNK*-deficient mice. We have corrected the text to fix this error.

b) The source data file for Figure 1C provides detailed information concerning the tumors and MIN lesions, including the age of the mice.

c) We examined tumors in Cre^+^ Control mice (*Cre^+^ Jnk1^+/+^ Jnk2^+/+^*) and Flox Control mice (*Cre^-^ Jnk1^LoxP/LoxP^ Jnk1^LoxP/LoxP^*) as well as experimental *JNK*-deficient mice (*Cre^+^ Jnk1^LoxP/LoxP^ Jnk1^LoxP/LoxP^*). All of these data are presented in the source data file. The Flox Control (*Cre^-^ Jnk1^LoxP/LoxP^ Jnk1^LoxP/LoxP^*) and experimental *JNK*-deficient mice (*Cre^+^ Jnk1^LoxP/LoxP^ Jnk1^LoxP/LoxP^*) are litter-mates, while the Cre^+^ Control mice were obtained from separate litters. For this reason, we present data using littermate control mice (*Cre^-^ Jnk1^LoxP/LoxP^ Jnk1^LoxP/LoxP^*) and experimental JNKdeficient mice (*Cre^+^ Jnk1^LoxP/LoxP^ Jnk1^LoxP/LoxP^*) in the Figure. Nevertheless, data from both Control strains and the experimental strain are presented in the source data file. No breast tumors were detected in the 22 Cre^+^ Control mice (*Cre^+^ Jnk1^+/+^ Jnk2^+/+^*) that were examined.

2) The main conclusion is that JNK suppresses breast cancer initiation; however the authors have recently published that JNK deletion results in a delay in mammary gland involution (Girnius et al., CDD). Therefore, could the lesions observed in Figure 7 be due to a lack of clearance post-weaning? This could lead to an expanded pool of cells susceptible to transformation and so a quirk of the WAP-Cre model. Have the authors considered using an alternative mammary-specific Cre driver such as K8-Cre or MMTV-Cre which would obviate the need to induce lactation? At the very least, this should be discussed in more detail. Can JNK be deleted in established tumors, either using an inhibitor or perhaps by way of an inducible knockdown orthotopic model?

We have revised the paper to cite our paper on the role of *JNK* in involution. We demonstrated that *JNK*-deficiency causes a delay in involution. However, at 14 days post-weaning, we found no differences in breast morphology between control and *JNK*-deficient mice. It is therefore not clear that suppressed involution contributes to the tumor phenotype of *JNK*-deficient mice. However, we have revised the Discussion to discuss this possibility. We did consider the use of *K8-Cre* and *MMTV-Cre* for these studies. However, the expression of *Cre* in these mouse lines is not restricted to the mammary epithelium. The use of these mice is therefore compromised by the ablation of *Jnk* genes in non-mammary tissues. The *WAP-Cre* model causes gene ablation that is restricted to the mammary epithelium.

3) There are 3 examples of ME-KO tumours in Figure 1D but the authors need to say if these are adenocarcinoma or adenosquamous carcinoma. Ideally at least two examples of each tumor type should be shown. It is not clear if the adenosquamous carcinomas originate in a JNK-deleted cell type since the WAP-cre driver is CK8-specific (Figure 1A) yet the adenosquamous lesions are predominately CK5 positive. Furthermore, there is no evidence for the trans-differentiation unless the authors can show these CK5+ tumours are JNK-deleted.

The three ME^KO^ tumors in Figure 1D are representative and include one adenocarcinoma, one adenosquamous carcinoma, and one mixed tumor with characteristics of both adenocarcinoma and adenosquamous carcinoma. We have revised the figure legend to make this clear. No *JNK* was detected by immunoblot analysis in the tumor cells isolated from the *JNK*-deficient mice.

We note that immunoblot analysis demonstrated no *JNK* expression in the adenosquamous carcinoma of *WAPCre^+^ Jnk1^LoxP/LoxP^ Jnk1^LoxP/LoxP^* mice (Figure 1D). This observation suggests that CK5+ adenosquamous carcinoma cells may arise by trans-differentiation of CK8 cells that express Cre, although we acknowledge that this conclusion is speculative. We acknowledge the speculative nature of this conclusion in the subsection “Disruption of JNK signaling causes breast cancer development”.

4) The RNAseq data presented in Figure 2 demonstrates an upregulation of WNT target genes in ME^KO^ cells, leading the authors to suggest that ME^KO^ tumors could exploit WNT signalling to drive proliferation. Validating this observation at the protein level through IHC of tumor tissue or Western blotting of ME^WT^ and ME^KO^ cell lines would add confidence to this proposed mechanism. Are WNT pathway components differentially expressed between JNK^WT^ and JNK^KO^ tumor cells in the p53mut background? Based on data in Figure 2E and 2F, would authors predict that JNK^KO^ tumor cells should be more sensitive to WNT pathway inhibitors? Has this been tested?

Our conclusions concerning the role of WNT signaling are based on the data presented in Figure 2 showing increased expression of *Wnt7b* and *Wnt10a* and increased expression of the WNT target genes *Axin2, Myc* and *Ccnd1*. This conclusion is based on gene expression analysis.

The reviewer requests that we provide protein-based evidence for WNT signaling in the tumor cells. A major signaling axis that mediates WNT function is the stabilization and nuclear accumulation of β-catenin that serves to increase WMT target gene expression. We therefore examined the nuclear accumulation of β-catenin by immunofluorescence analysis. Figure A shows accumulation of β-catenin in the nucleus of JNK-deficient tumor cells. This observation is fully consistent with our data showing that tumor cells from *JNK* knockout mice exhibit increased expression of WNT target genes.

We tested the role of autocrine Wnt signaling in the cultured tumour cells by examining the effect of the addition of the WNT antagonist DICKKOPF (R&D Systems) to the culture medium. We found that DICKKOPF caused no change in tumor cell proliferation. We omitted these data from the study for two reasons. First, it is not clear that the rate of proliferation is the most important in vitro property of tumour cells. Second, it is not clear that WNT inhibition by DICKKOPFin vitro mimics the effect of WNT signaling inhibition during tumor development in vivo. We believe that substantially more work is required to establish the role of WNT signaling in the observed tumor development and that this is beyond the scope of the current study.

**Author response image 1. respfig1:** Mammospheres prepared from Control (left panel) and *JNK*-deficient (right panel) tumor cells were embedded in paraffin, sectioned, and stained with DAPI and an antibody to β-catenin. We found that β-catenin was restricted to the cytoplasm of Control tumor cells, but β-catenin was detected in both the nucleus and cytoplasm of *JNK*-deficient tumor cells.

5) The detection of genomic instability indicators in ME^KO^ tumor cell lines is interesting but perhaps expected when comparing tumor cells with normal epithelial cells. It would have been useful to compare ME^KO^ with ME^WT^ tumor cell lines. It does looks like in the p53mut background, JNK^KO^ cells have less SNVs and indels than JNK^WT^ cells (Figure 1—figure supplement 2A). Authors should comment on this.

While the number of SNV and indels caused by TRP53 deficiency was greater than that caused by *JNK* deficiency, the number of CNV were similar on both genetic backgrounds. These observations indicate that while both TRP53deficiency and *JNK*-deficiency cause genomic instability, the mechanisms of genomic instability caused by these two pathways is different. We have added additional discussion of this point to the text.

6) Do gene expression changes associated with JNK deficiency in mammary tumor cells correlate with previously published gene signatures associated with genomic instability or DNA damage?

Gene set enrichment analysis demonstrates decreased expression of a “DNA Repair” gene signature by *JNK* deficient tumor cells (Figure 2—figure supplement 1). These data are included in the revised manuscript.

Points for clarification/discussion:

7) Human breast cancers retain expression of JNK1 and JNK2 but exhibit inactivating mutations in upstream components, which will strongly impair activation of JNK. This is obviously not the same as the complete ablation of JNK expression that the authors employ in this model. Whilst the data presented are strong and provide important new insights into the role of JNK in this murine model, the authors do need to explicitly acknowledge and comment on this limitation of their model.

We agree – a clear goal for future studies will be to directly test the role of MEKK1 and MAP2K4 *loss-of-function* mutations in breast cancer. We have revised the Discussion to state this important point.

8) At several places in the manuscript (Introduction and Conclusion) the authors state that their discovery that loss of JNK1/2 expression promotes breast cancer may 'present an opportunity for therapeutic intervention'. It is not immediately obvious how a small molecule could rescue the effects of losing JNK1/2 expression to prevent genome instability. Presumably they are thinking about a disease stratification approach in which those breast cancers with loss of function mutations on the JNK pathway and enhanced genome stability may be more susceptible to intervention with drugs targeting DNA Damage Response (DDR) pathways, analogous to synthetic lethality that has helped to drive approval of PARPi? Or do they envisage combination of JNKi and DDRi as being synthetic lethal in cells that do not have JNK pathway mutations? A little more information here, or discussion, would help to clarify ideas and potential significance for the reader.

We agree that the conclusions on this point were not stated clearly in the original version of our manuscript. Our expectation is that *JNK*-deficiency would exhibit synthetic lethality with drugs that cause DNA damage or prevent DNA repair. We have revised the Discussion to state this more clearly.

9) In Figure 3 and Figure 3—figure supplement 2 the authors describe how the Kras locus is more frequently amplified in JNK^WT^ mice compared to JNK^KO^. Could the authors comment on the significance of this observation? Are they proposing that because the JNK^KO^ tumours display increased genome instability they are likely to acquire many more oncogenic events and thus become less reliant on the amplification of the Kras locus which is prevalent in the JNK^WT^ mice?

We agree that our conclusions were not clearly stated in the original manuscript. As stated by the reviewers, we suspect that the *Kras* amplification detected in WT tumors was functionally substituted by the mutation of other oncogenic pathways in the *JNK* knockout tumors. We have revised the last paragraph of the subsection “JNK deficiency rapidly accelerates tumor development in a mouse model of breast cancer” to state this more clearly and by including gene set enrichment analysis for a *Kras* signaling signature (Figure 3—figure supplement 3).